# SERS nose arrays based on a signal differentiation approach for TNT gas detection
Peitao Dong [1,8], Haiyang Yang[2,8], Tianran Wang [1,3] ✉, Siyue Xiong[1], Li Kuang[2], Weihong Qi[4], Xiaohua Chen[5], Lixia Yang[6], Qiuyun Fan[7], Dingbang Xiao [1] & Xuezhong Wu[1]

TNT, a well-known explosive, is highly toxic and difficult to decompose, making the detection of trace amounts of residual TNT in the environment a topic of significant research importance. Label-free surface-enhanced Raman spectroscopy (SERS) has been demonstrated to be capable of capturing rich compositional information from the sample being tested. Here we show a SERS nose array that contains six individual SERS substrates composed of different components based on a signal differentiation approach (SD-SERS arrays). In this strategy, the SD-SERS arrays integrate differentiated signal structures, physically enhanced structures, and structures with varied adsorption capabilities. Through the differentiated information obtained from SD-SERS arrays, further integration with machine learning algorithms demonstrates the high accuracy of SD-SERS arrays in classifying TNT and structurally similar 2,4-DNPA, as well as in distinguishing between gases at different concentrations. The SERS nose based on SD-SERS arrays presents a convenient and broadly applicable technology with great potential for substance classification and concentration categorization.

TNT is a highly regarded explosive due to its powerful explosive energy and stable performance. TNT has been found to have a straightforward synthesis process and high energy output. This has resulted in the substance being utilized extensively in a number of fields, including chemical engineering, coal mining, and military operations. It remains the most extensively produced and used explosive, as well as a frequent substance involved in explosive-related incidents, posing serious threats to modern life, national security, and social stability. TNT is also highly toxic and resistant to degradation, allowing it to persist in the environment for decades. The long-term environmental contamination by TNT poses significant risks to human health and ecosystems. As a known carcinogen, TNT can enter the human body through the respiratory tract, leading to immune system dysfunction and severe health issues. Consequently, the detection of trace amounts of residual TNT in the environment is of critical research importance.

Common methods for detecting TNT include gas chromatography[1], liquid chromatography[2], mass spectrometry[3], and terahertz wave technology[4]. Whilst these techniques offer high sensitivity, they are often limited by bulky equipment, complex operational processes, and high costs, making them unsuitable for portable detection applications. Therefore, there is an urgent need for a detection method that combines high sensitivity and trace-level detection capabilities with portability and low cost[5]. To address this challenge, researchers have drawn inspiration from the exceptional sensitivity and discriminatory abilities of biological olfactory systems. By studying and mimicking biological olfaction, they have leveraged the differential responses of numerous "neurons" to various odors to achieve highly sensitive odor detection and discrimination. This has led to the development of biomimetic "nose" sensor technology, enabling the creation of advanced olfactory sensors and driving progress in biochemical sensing technologies. The biomimetic nose employs an array of multiple sensors to form an artificial olfactory system capable of accurately identifying and distinguishing target analytes. This technology offers innovative solutions for applications, including battlefield reconnaissance, medical diagnostics, food quality inspection, and counter-terrorism security.

[1]College of Intelligence Science and Technology, National University of Defense Technology, Changsha, Hunan, China. [2]School of Computer Science and Engineering, Central South University, Changsha, Hunan, China. [3]School of Intelligent Manufacturing and Electronic Engineering, Wenzhou University of Technology, Wenzhou, Zhejiang, China. [4]State Key Laboratory of Solidification Processing and Center of Advanced Lubrication and Seal Materials, Northwestern Polytechnical University, Xi'an, China. [5]Department of Laboratory Medicine, General Hospital of Central Theater Command, Wuhan, Hubei, China. [6]Changsha Institute for Food and Drug Control, Changsha, Hunan, China. [7]Hunan Changsha Ecological and Environmental Monitoring Center, Changsha, Hunan, China. [8]These authors contributed equally: Peitao Dong, Haiyang Yang. ✉e-mail: wangtianran@wzut.edu.cn

Biomimetic nose sensors are principally through the measurement of changes in fluorescence intensity, electrical signal responses, or mass variations within sensor arrays to achieve detection. For nitro-explosives, researchers have developed various biomimetic nose sensor arrays[6–9], including chemiresistive sensors[10–12], colorimetric[13,14] and also optical absorption sensor arrays[15]. Yang et al. designed an electronic nose based on UV absorption spectroscopy[15]. This system employed a sensor array composed of nine sensors created from varying ratios of $SnO_2/WO_3$, enabling the detection of nitro-explosives such as trinitrotoluene (TNT), pentaerythritol tetranitrate (PETN), and cyclotetramethylene tetranitroamine (HMX), with detection limits as low as 500 ng, 100 ng, and 50 ng, respectively. The system further utilized a CNN-LSTM model for rapid and accurate classification of the detected substances. Qiu et al.[14] developed molecularly imprinted amorphous photonic crystals (MIAPCs). These arrays enabled the detection of nitro-explosives by analyzing color changes in the sensor arrays, combined with pattern recognition techniques, to identify substances such as 2,4,6-trinitrotoluene (TNT), 1,3,5-trinitro-1,3,5-triazinane (RDX), and octahydro-1,3,5,7-tetranitro-1,3,5,7-tetrazocine (HMX). Che et al.[16] presented a sensor array unit that has the capacity to measure both fluorescence and optical stability in a single experiment. This advanced sensor demonstrated superior multi-target discrimination capabilities, enabling stable and multiplex detection of hazardous chemicals, including explosives.

Despite significant advancements in the development of sensor arrays for trace-level detection of explosives, the extremely low saturated vapor pressures of explosives and their dispersion in the environment remain major challenges for the practical application of electronic noses. These low concentrations demand sensor arrays with ultra-low detection limits. However, traditional biomimetic noses often face limitations in selectivity and stability. This is primarily due to their reliance on one-dimensional signal outputs, such as intensity, which fail to provide detailed information about the structure or composition of the detected substances. Consequently, these limitations impede the efficacy of biomimetic nose sensors, curtailing the scope of detectable substances and diminishing the precision of detection in practical applications.

Surface-enhanced Raman spectroscopy (SERS) is an analytical technique that enhances Raman scattering signals by adsorbing target molecules onto a SERS substrate. This amplification enables the extraction of structural information about target molecules, offering high sensitivity, high spatial resolution, rapid response, and broad applicability in trace detection[17–21]. SERS shows great potential in applications such as food safety[22], environmental pollutant monitoring[23], protein 3D structure determination[24], and disease diagnosis[25]. Notwithstanding these advances, challenges remain when detecting structurally similar or complex substances. The identification of substances can be hindered by issues such as the overlap of spectra and the complexity of spectral classification. For instance, in the detection of nitro-explosives such as TNT using a singular SERS substrate, the presence of structurally analogous compounds, including 2,4-dinitrophenylacetic acid (2,4-DNPA) and 4-nitrotoluene (4-NT), can lead to erroneous identifications. The limitations of traditional electronic noses can be addressed by biomimetic noses based on SERS, which provide structural information about detected substances. By leveraging the differential signal intensities generated by multiple SERS substrates, this approach delivers multidimensional spectral information, reducing misidentification caused by interference from structurally similar compounds during detection.

To achieve the outlined objectives, constructing a sensor array of SERS substrates, referred to as a "SERS nose," is essential[26,27]. We propose a Signal-Differentiated SERS (SD-SERS) substrate array that contains six individual SERS substrates as the sensing component of the "SERS nose." In the SD-SERS array, detection signals are influenced by three key factors: the one is the chemical enhancement (CM) effect, arising from charge transfer interactions between different substrates and target molecules. The two are the adsorption capability of substrates, modulated by surface-modified self-assembled monolayers. The electromagnetic (EM) enhancement is the last one, which can enhance the Raman signal intensity. These factors generate differentiated signal intensities, enabling enhanced detection performance. We selected two potential 2D MXene materials with intrinsic chemical enhancement properties, $Mo_2C$ MXene[28] and $Ti_3C_2$ MXene[29,30], as supporting materials to provide differentiation in chemical enhancement for the SD-SERS array. Additionally, three chemical molecules with varying adsorption affinities for TNT were used to modify the substrate surfaces, introducing adsorption capability differentiation. Given TNT's extremely low saturated vapor pressure and its diffusion in the environment, resulting in low ambient concentrations, we further enhanced the detection limits of the SD-SERS array by integrating gold nanobipyramids (AuNBPs) onto the surfaces of the 2D materials as EM enhancement structures.

This study demonstrates that the SD-SERS array enhances TNT gas adsorption in the environment through surface modification with self-assembled monolayers (SML) on the substrates. Trace detection of TNT is achieved via EM enhancement from SERS hotspots. For structurally similar substances, differences in adsorption and chemical enhancement within the SD-SERS array amplify signal distinctions. By analyzing spectral data with machine learning algorithms, the system successfully classified TNT and its structural analogs (2,4-DNPA). These findings highlight the potential of the SD-SERS sensor array-based "SERS nose" for trace detection and differentiation of structurally similar compounds.

## Results and discussion

To achieve improved detection performance, the EM enhancement structure of the SD-SERS arrays exhibits superior electromagnetic enhancement hotspots. Due to the influence of electromagnetic field distribution, electromagnetic enhancement 'hotspots' are subject to the tip effect. To this end, we compared the electromagnetic enhancement performance of three nanostructures: gold nanostars (AuNSs), gold nanorods (AuNRs), and AuNBPs. In this study, the electromagnetic enhancement signals generated at hotspots formed by head-to-head configurations of these nanostructures were calculated using the finite-difference time-domain (FDTD) method. Supplementary Fig. 1s illustrates the tip-to-tip hotspots for the three types of nanostructure, with a tip-to-tip distance of 1.5 nm in all three cases. Based on the simulation results shown in Supplementary Fig. 2s, the electromagnetic field distributions at the tip-to-tip regions of these structures reveal strong enhanced hotspots in the nanogaps. The intensity of these physical hotspots depends on the value of $E/E_0$. Since the enhancement factor is approximately proportional to the fourth power of $E/E_0$, a comparison of the $E/E_0$ values among the three nanostructures indicates that AuNBPs exhibit superior electromagnetic enhancement performance. To enable the detection of trace molecules in a gaseous environment, this study employed AuNBPs, which offer stronger EM enhancement properties, were employed to amplify the detected Raman signals.

Figure 1 summarizes the preparation process of the SD-SERS array. Figure 1a–c details the synthesis of AuNBPs using a seed-mediated method. First, the gold precursor is reduced by sodium borohydride in a solution containing CTAC and sodium citrate (Fig. 1a). Under the influence of surfactants, the reduced gold atoms nucleate and grow into decahedral gold seeds (Fig. 1b). The seed solution is then prepared by reducing a chloroauric acid precursor (adjusted with HCl) using ascorbic acid (AA) and mixing it with the gold seed solution. In the growth solution, silver nitrate and CTAB facilitate the deposition of gold atoms around the gold seed core. $Ag^+$ reacts with $Br^-$ from CTAB to reduce the repulsive forces between the surfactant molecules on the seed surface and allow gold atom deposition. Additionally, the formation of silver bromide adsorbs preferentially onto specific crystal facets, directing particle growth. This process results in the formation of AuNBPs, as shown in Fig. 1c. Due to the presence of surfactants, the surface of AuNBPs synthesized via the seed-mediated growth method has a positive charge (Fig. 1d). In contrast, freshly prepared MXene materials have negatively charged surface functional groups. The electrostatic attraction[31] between these oppositely charged surfaces, combined with the high formation energy of AuNBPs, promotes the attachment between MXene and AuNBPs. Using this approach, two heterostructures—$Ti_3C_2$ MXene-

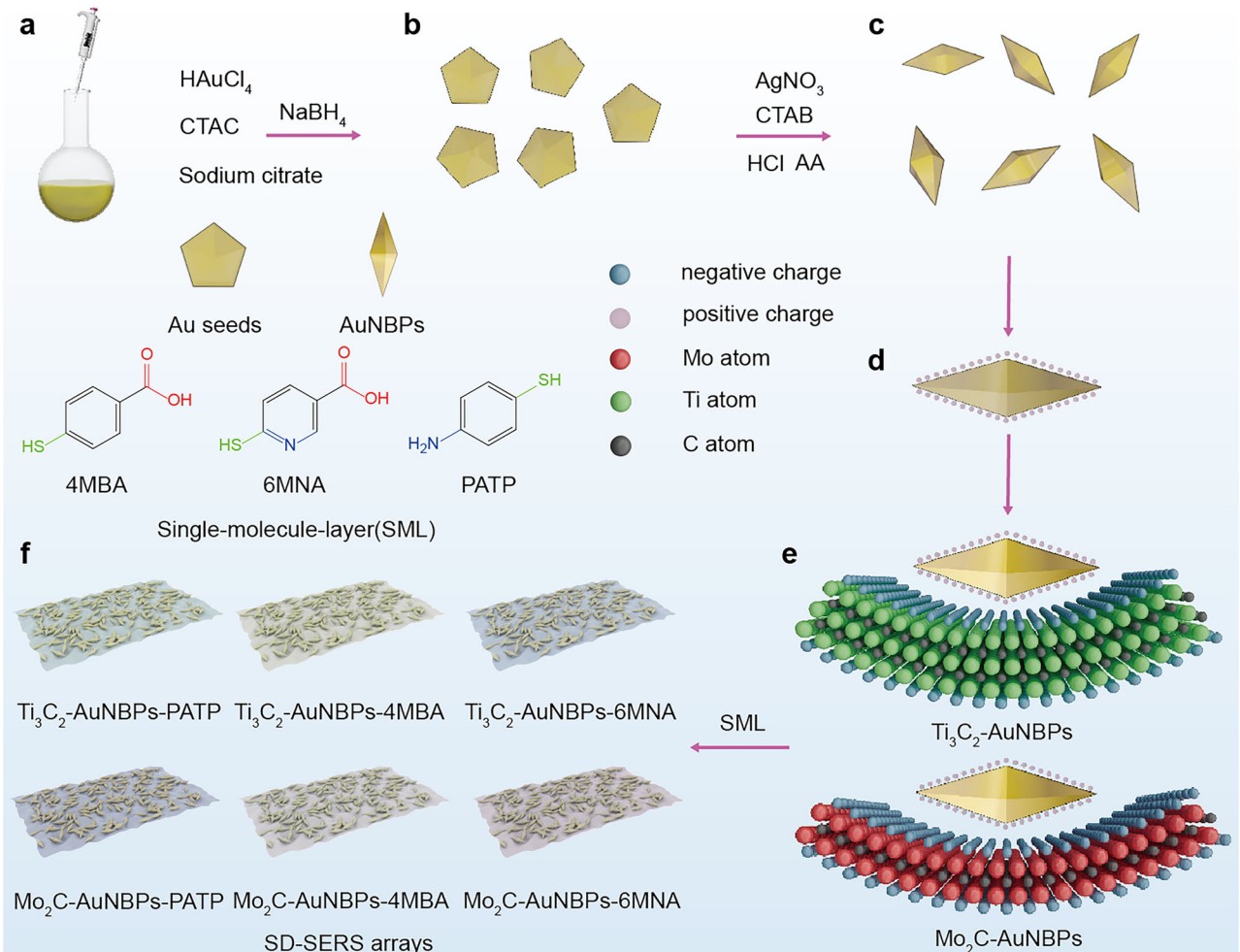

**Fig. 1 | Schematic diagrams illustrating the preparations of the SD-SERS arrays.** **a** Preparation of the reaction solution for AuNBPs seeds. The prepared (**b**) decahedral gold seeds and (**c**) AuNBPs. **d** The surface of AuNBPs carrie positive charge. **e** Two types of heterostructures, Ti$_3$C$_2$ MXene -AuNBPs and Mo$_2$C MXene -AuNBPs, were prepared using a self-assembly method assisted by electrostatic attraction method. **f** By self-assembling three types of monolayers on the surfaces of the two structures, six SERS substrates with similar EM enhancement structure, different adsorption capacities, and chemical enhancement properties were obtained. These six substrates collectively form the SD-SERS arrays.

AuNBPs and Mo$_2$C MXene-AuNBPs—were successfully fabricated (Fig. 1e). To create a SERS nose array with varied adsorption capabilities, three self-assembled monolayers (SMLs: 4MBA, 6MNA, and PATP) were grafted onto the surfaces of the Ti$_3$C$_2$ MXene-AuNBPs and Mo$_2$C MXene-AuNBPs structures. These SMLs interact differently with target analytes, imparting distinct adsorption properties to each structure and resulting in six unique SERS substrates. These substrates form the SD-SERS array, as shown in Fig. 1f.

Figure 2a(i) shows the morphology of the synthesized gold seeds, which have an average size of 15.3 ± 0.13 nm, as analyzed in Supplementary Fig. 3s. The magnified image in Fig. 2a(ii) reveals a decahedral structure with five twinned crystal planes. This morphology was achieved by using CTAC as a surfactant and extending the seed maturation time, which significantly enhances the yield of AuNBPs in subsequent synthesis steps. Figure 2b(i), (ii) illustrate the morphology of the synthesized AuNBPs. During the growth process, Ag$^+$ ions play a critical role in the formation of anisotropic gold nanocrystals. The Ag$^+$-assisted growth mechanism relies on two primary hypotheses: underpotential deposition and the selective adsorption of silver bromide on specific gold crystal facets. Due to the surface energy hierarchy of gold crystal facets ([110] > [100] > [111]), silver atoms preferentially deposit on high-energy facets, suppressing growth in those directions[32]. Under CTAB conditions, gold atoms preferentially grow on the [111] facets, resulting in the uniform AuNBP morphology observed in

Fig. 2b. Statistical analysis in Fig. 2c shows that the average size of the synthesized AuNBPs is 53.6 ± 0.17 nm.

Tuning the absorption peak of the substrate to match the wavelength of the incident laser is critical for achieving plasmonic resonance for SERS performance. In this study, the Raman detection system employs an excitation wavelength of 785 nm. To optimize the Raman signal, the longitudinal plasmon resonance absorption peak of the AuNBPs was tuned to approximately 785 nm. By keeping other parameters constant and varying the gold seed content, AuNBPs with different absorption peaks were synthesized. Supplementary Fig. 4s displays the morphologies of the AuNBPs synthesized with varying seed contents. The results show that as the gold seed content decreases, the length of the AuNBPs increases. Supplementary Fig. 5s illustrates the absorption spectra of AuNBPs synthesized using gold seed solution volumes of 70 μL, 90 μL, 110 μL, and 130 μL. When the seed content is set to 70 μL, the resulting product exhibits an absorption peak near 788 nm, which closely aligns with the excitation wavelength. Consequently, the AuNBPs synthesized with this seed content were chosen as the signal-enhancing structures for the SD-SERS array.

During the self-assembly process between MXene materials and AuNBPs, the positive surface charge of AuNBPs interacts electrostatically with the negatively charged MXene surfaces, facilitating their connection. The surface potentials of Ti$_3$C$_2$ MXene and Mo$_2$C MXenes, as well as that of AuNBPs, were characterized at Fig. 2d. For Ti$_3$C$_2$ MXene and Mo$_2$C

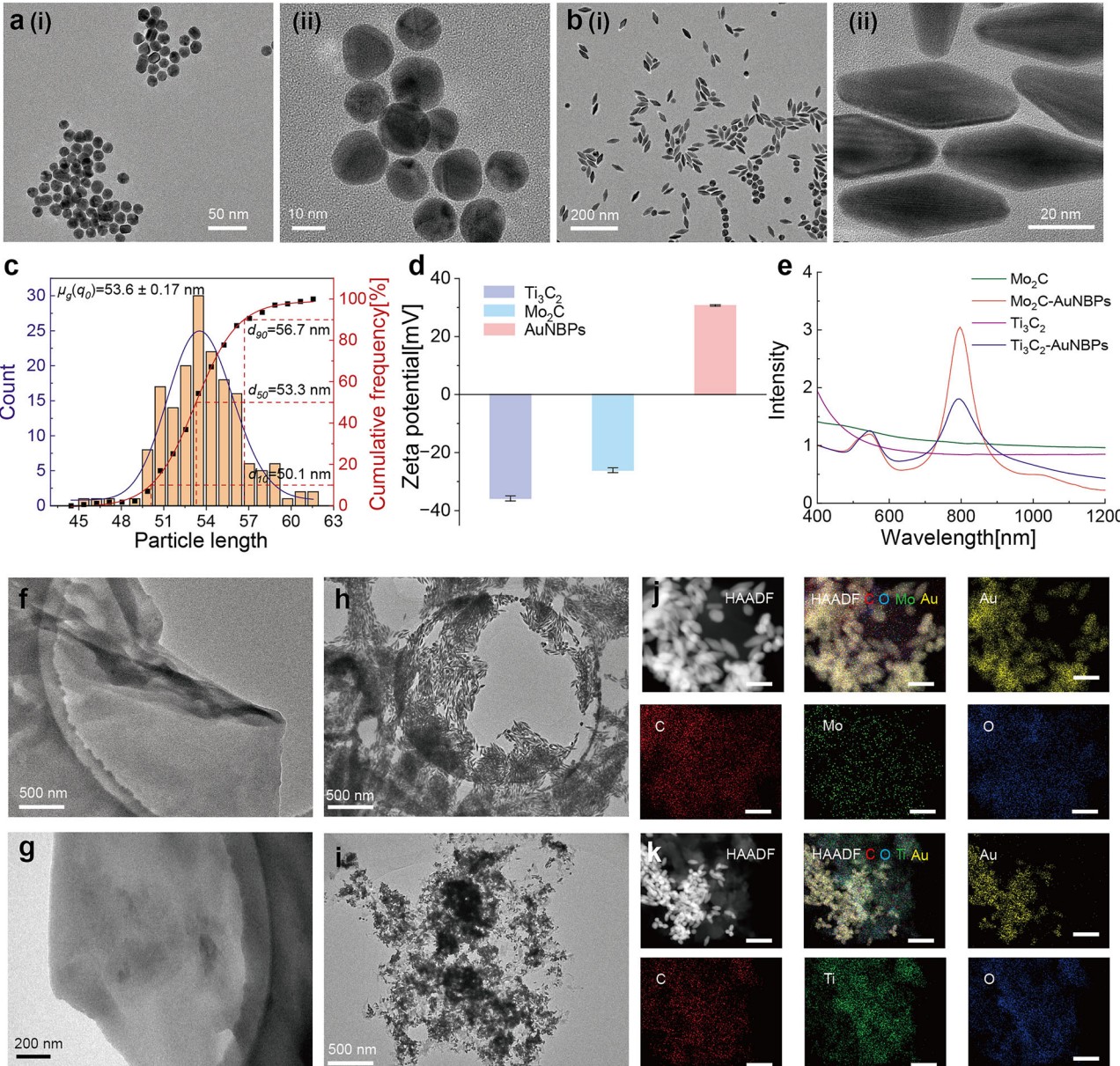

**Fig. 2 | Characterizations of the MXene-AuNBPs.** The transmission electron microscopy (TEM) images of (**a**) Au seeds with different magnifications and (**b**) AuNBPs with different magnifications. **c** Particle size distribution diagram of the AuNBPs. **d**. The surface potential of different structures. **e** Absorption spectra of the four structures.The transmission electron microscopy (TEM) images of (**f**) Mo$_2$C MXene, (**g**) Ti$_3$C$_2$ MXenes, (**h**) Mo$_2$C MXene -AuNBPs, and (**i**) Ti$_3$C$_2$ MXene -AuNBPs. HAADF images and the corresponding EDS elemental maps of **j**. Mo$_2$C MXene -AuNBPs with Au, Mo, C, and O. HAADF images and the corresponding EDS elemental maps of (**k**) Ti$_3$C$_2$ MXene -AuNBPs with Au, Ti, C, and O.

MXene, the surface potential is $-35.8 \pm 0.87$ mV with detailed data shown in Supplementary Fig. 6s(a) and $-26.1 \pm 0.83$ mV with detailed data shown in Supplementary Fig. 6s(b), respectively. Supplementary Table 1s and table 2s provide the peak and area information for the zeta potential of Ti$_3$C$_2$ MXene and Mo$_2$C MXene. These results confirm that both MXene surfaces carry negative charges due to the presence of -OH and -COOH functional groups. The surface potential of AuNBPs is $30.7 \pm 0.25$ mV, as illustrated in Supplementary Fig. 6s(c), with Supplementary Table 3s detailing the positions of the three primary peaks in the zeta potential spectrum. This positive surface charge is attributed to the presence of the CTAB surfactant on the AuNBPs surface. Additionally, the high formation energy of AuNBPs contributes to increased adhesion energy, further facilitating their self-assembly with MXene materials. The UV-vis spectroscopy is further used to track the formation of the heterostructures. As shown in Fig. 2e, both MXene structures alone exhibit no significant absorption peaks. However,

after the self-assembly of AuNBPs with the MXene structures, the resulting heterostructures display absorption peaks near 788 nm, corresponding to the longitudinal surface plasmon resonance (SPR) peak of AuNBPs. This observation further confirms the successful formation of the MXene-AuNBPs heterostructures.

The transmission electron microscopy (TEM) images in Fig. 2f, g depict the structures of few-layer Mo$_2$C MXenes and Ti$_3$C$_2$ MXenes, respectively. Figure 2h shows the Mo$_2$C-AuNBPs heterostructure formed by the connection of Mo$_2$C MXene and AuNBPs. The TEM and SEM images of the heterostructure at different magnifications are provided in Supplementary Fig. 7s. These images demonstrate that AuNBPs are uniformly attached to the two-dimensional Mo$_2$C MXene surface. Moreover, the synthesized Mo$_2$C MXene-AuNBPs exhibits minimal aggregation of the AuNBPs. The closely packed arrangement of AuNBPs within the synthesized SERS substrate provides numerous electromagnetic enhancement

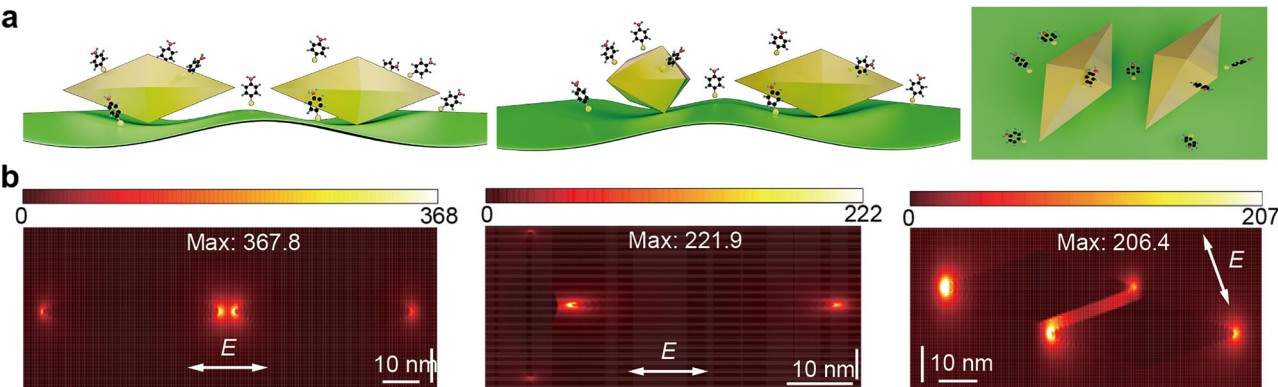

**Fig. 3 | 'hotspots' between two AuNBPs. a** The interactions between detection molecules and the MXene-AuNBPs under different assembly configurations of AuNBPs.
**b** The electromagnetic field distribution of AuNBPs with three kinds of contact modes.

'hotspots' for Mo$_2$C MXene-AuNBPs. Figure 2j presents HAADF images and the corresponding EDS elemental maps for C, O, Mo, and Au, confirming the successful connection of Mo$_2$C MXene with AuNBPs. Similarly, Fig. 2i shows the Ti$_3$C$_2$ MXene-AuNBPs heterostructure, with the formation of the heterostructure supported by HAADF images and EDS elemental maps for C, O, Ti, and Au in Fig. 2k.

The SERS performance of the synthesized MXene-AuNBP was evaluated using two common Raman reporter molecules. These heterostructures leverage the electromagnetic enhancement effects of AuNBPs and the chemical enhancement effects of MXenes. Figure 3a illustrates the interactions between the detection molecules and the MXene-AuNBP under different AuNBPs assembly configurations. When the molecules adsorb onto the AuNBP surface, the localized plasmon resonance effect of AuNBPs generates a strong electromagnetic field, amplifying the Raman signal. As shown in Fig. 3a, electromagnetic 'hotspots' are formed between AuNBPs in various contact configurations, including head-to-head, head-to-shoulder, and shoulder-to-shoulder arrangements. Figure 3b presents the electromagnetic field distribution of AuNBPs under these configurations, as calculated through FDTD simulations. Compared to excitation perpendicular to the contact direction (see Supplementary Fig. 8s), parallel excitation along the polarization direction of the AuNBP contact surfaces reveals a significant polarization-dependent enhancement. This enhancement produces electromagnetic hotspots with maximum E/E$_0$ values of 367.8, 221.9, and 206.4 for head-to-head, head-to-shoulder, and shoulder-to-shoulder configurations, respectively. Since the enhancement factor is approximately proportional to the fourth power of (E/E$_0$), it is evident that the physical hotspots formed by AuNBPs substantially amplify the Raman signal.

Further investigations were conducted using density functional theory (DFT) to study the charge transfer between two types of MXenes (Ti$_3$C$_2$ MXenes and Mo$_2$C MXenes) and the adsorbed molecules 4MBA and PATP. When the detection molecules adsorb onto the MXene surface, charge transfer between the metal atoms on MXene and the molecules enhances the Raman signal via chemical enhancement. To evaluate the charge transfer between 2D materials and adsorbed molecules, we only use Ti$_3$C$_2$ and Mo$_2$C rather than Ti$_3$C$_2$ MXenes and Mo$_2$C MXene. Supplementary Fig. 9s and Fig. 4a illustrate the optimized structural diagrams and charge density differences for the Mo$_2$C/PATP, Ti$_3$C$_2$/PATP, Mo$_2$C/4MBA, and Ti$_3$C$_2$/4MBA configurations. In these images, the red regions represent electron accumulation and the green regions denote electron depletion. The results reveal that the MXenes lose electrons, while the adsorbed molecules gain electrons, particularly in the bonding regions and on the molecular surfaces. This demonstrates that in the adsorption structures, MXenes primarily act as electron donors, transferring electrons to the adsorbed molecules.

The adsorption energy E$_{ads}$ for these configurations was calculated using the formula: E$_{ads}$= E$_{molecule+surface}$ -E$_{surface}$ − E$_{molecule}$. The adsorption

energies for Mo$_2$C/PATP, Ti$_3$C$_2$/PATP, Mo$_2$C/4MBA, and Ti$_3$C$_2$/4MBA were found to be −5.92, −7.43, −5.47, and −5.63 eV, respectively. Since a lower energy indicates higher stability, these results suggest that the adsorption of chemical molecules onto the surfaces of both 2D materials is energetically favorable and stable. When the adsorption energy is low, charge transfer between the adsorbate and a greater amount of charge may be transferred[33,34]. Figure 4b quantifies the amount of charge transfer for each MXene after adsorbing the molecules. For the same adsorbed molecule (4MBA), the Ti$_3$C$_2$/4MBA system exhibits stronger charge transfer compared to Mo$_2$C/4MBA. This indicates that Ti$_3$C$_2$ facilitates greater charge transfer to both 4MBA and PATP molecules. In the context of Raman detection, this translates to stronger chemical enhancement and superior SERS performance. Similarly, for PATP, Ti$_3$C$_2$/PATP demonstrates a greater extent of charge transfer. Theoretically, this implies that the Ti$_3$C$_2$/AuNBPs offers stronger enhancement capabilities when detecting these two molecules compared to Mo$_2$C/AuNBPs. This differentiated enhancement mechanism provides the SD-SERS array with a more diverse Raman signal intensity dimension, enabling improved differentiation of overlapping spectra in subsequent detection processes.

To validate the theoretical predictions, the Raman spectra of AuNBPs and MXene were measured before and after the adsorption of the two detection molecules-PATP and 4MBA. Figure 4c shows the Raman detection results for 10$^{-2}$M PATP and 10$^{-2}$M 4MBA using two MXene substrates. The characteristic Raman peaks of PATP and 4MBA, with the 1078 cm$^{-1}$ and 1587 cm$^{-1}$ peaks corresponding to C-C and C-S stretching vibrations and the stretching vibrations of the benzene ring's C-C bonds[35]. As shown in Fig. 4c, no PATP and 4MBA Raman peak signals appear on bare Mo$_2$C MXene and Ti$_3$C$_2$ MXene. Similarly, when detecting 10$^{-2}$M PATP and 10$^{-2}$M 4MBA using no SERS substrate, only PATP peak can be observed on the figure. Where two MXene substrates were used to detect 10$^{-2}$M PATP and 10$^{-2}$M 4MBA, all Raman signals at characteristic peak were enhanced. It can be concluded that the two types of MXenes can enhance the Raman signals of adsorbed molecules through chemical enhancement mechanisms, which is consistent with the computational results. To further assess the SERS performance of MXene-AuNBPs, Fig. 4d compares the SERS performances for detection of PATP of bare MXene, AuNBPs, and MXene-AuNBPs. A strong Raman signal for PATP at 10$^{-5}$ M was observed when the MXene surface was coupled with AuNBPs to form MXene-AuNBPs, indicating a significant enhancement in SERS performance after coupling with AuNBPs. Comparing the detection of 10$^{-5}$ M PATP using AuNBPs and MXene-AuNBPs, the Raman signal from MXene-AuNBPs is also stronger. This enhanced signal can be attributed to two main factors: (1) The chemical enhancement capability of MXene boosts the intrinsic SERS performance of the heterostructure, surpassing the enhancement provided by AuNBPs alone. (2) MXene's larger specific surface area enables it to adsorb more PATP molecules, and these molecules are positioned near the "hotspots" formed by AuNBPs (as illustrated in Fig. 3a). In contrast, PATP

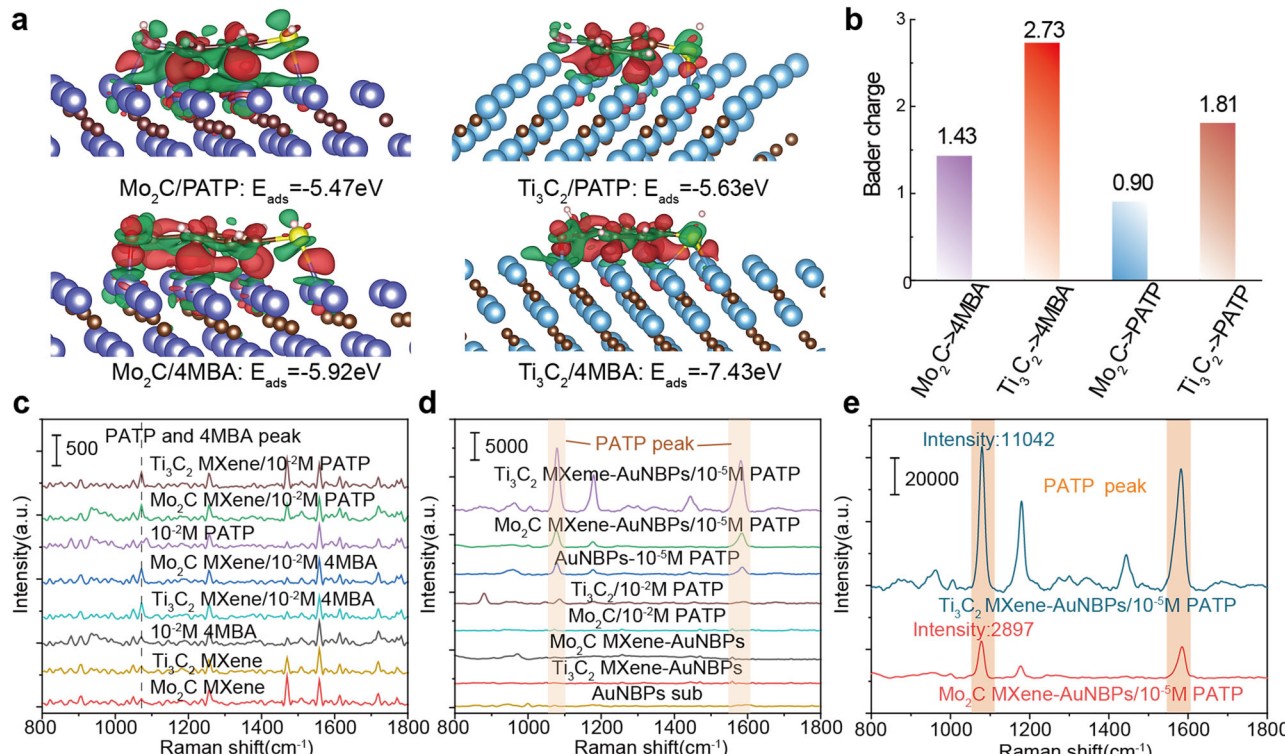

**Fig. 4 | Illustration of signal differentiation approach for MXene-AuNBPs based on charge transfer differences. a** Charge density difference, adsorption energies, and **b** the amount of charge transfer of four adsorption models. The adsorption energies for Mo₂C/PATP, Ti₃C₂/PATP, Mo₂C/4MBA, and Ti₃C₂/4MBA were found to be −5.92, −7.43, −5.47, and −5.63 eV, respectively. The Raman spectra of PATP and 4MBA detected by **c.** bare MXenes and detection of PATP by AuNBPs and MXenes-AuNBPs. **d.** The SERS performances for detection of PATP of bare MXene, AuNBPs, and MXene-AuNBPs. **e.** The details of Raman spectra for $10^{-5}$ M PATP using Ti₃C₂ MXen -AuNBPs and Mo₂C MXenex-AuNBPs heterostructures.

molecules adsorbed directly onto the AuNBPs substrate are fewer, and the likelihood of these molecules being located within the "hotspots" is lower compared to the MXene-AuNBP. To verify the differences in chemical enhancement provided by different MXenes, Fig. 4e compares the Raman detection of $10^{-5}$ M PATP using Ti₃C₂ MXene -AuNBPs and Mo₂C MXene-AuNBPs. The results show that Ti₃C₂ MXene-AuNBPs produce a stronger Raman signal than Mo₂C MXene-AuNBPs, indicating superior SERS performance of Ti₃C₂ MXen-AuNBPs for PATP detection. These findings are consistent with the DFT calculations and theoretical predictions. This indicates that the MXene-AuNBPs synthesized from different materials can exhibit varied enhancement effects when detecting different substances. The results confirm the feasibility of utilizing charge-transfer-based differentiated signal enhancement for detection applications.

Figure 5a illustrates the method for detecting TNT and nitro compounds in gaseous form using SD-SERS arrays. Figure 5a(i) shows the six SERS substrates that make up the SD-SERS arrays. These substrates are composed of two composite structures—Ti₃C₂ MXene-AuNBPs and Mo₂C MXene-AuNBPs—modified with three types of monolayer capture molecules that interact with nitro compounds, as shown in Fig. 5a(ii). During the capture of these gaseous molecules, the capture molecules enrich nitro compounds on the surfaces of the MXene and AuNBPs structures. As depicted in Fig. 5a(iii), the interaction between nitro compounds and MXene leads to charge transfer, providing chemical enhancement to the Raman signal. Simultaneously, these molecules positioned in the 'hotspots' created by the AuNBPs experience stronger electromagnetic enhancement of the Raman signal.

The monolayer capture molecules consist of three substances: PATP[36], 4MBA[37], and 6MNA[38]. Their structures are shown in the inset of Fig. 1. These molecules interact with nitro compounds, enabling their capture. Taking PATP as an example, it can interact with TNT through three primary mechanisms to achieve TNT capture, as illustrated in Figure 5b[39,40]. (1)

Meisenheimer complex formation: PATP forms a Meisenheimer complex with TNT through specific chemical interactions. (2) TNT anions and PATP cations interact via electrostatic ion-pair interactions. (3) π-π Stacking Interaction: TNT acts as a π-acceptor while PATP serves as a π-donor, enabling π-π stacking interactions. For 4MBA, its carboxylic acid group interacts with TNT's nitro group through hydrogen bonding, forming a Lewis acid-base pair. Similarly, 6MNA shares structural similarities with 4MBA and interacts with TNT in a similar manner. The single molecular layer used in this study reacts with the electron-accepting nitro groups of nitro compounds through the aromatic ring, amino, and carboxylic acid groups. In theory, most of the active sites of the capture molecules modified on the MXene-AuNBPs' surface can be used for TNT adsorption.

To validate that this interaction enhances the adsorption of more compounds onto the SERS substrate, thereby improving the detected signal, this study utilizes Mo₂C MXene-AuNBPs to investigate the effect of modifying the substrate with a monolayer of capture molecules on the detection of TNT. Supplementary Fig. 10s shows the intrinsic Raman spectrum of TNT powder, with its main characteristic peak appearing at 1359 cm⁻¹. Figure 5c(i) presents the Raman signals of TNT after 1 hour adsorption of 67 ppb TNT gas, comparing substrates modified with $10^{-6}$ M PATP and unmodified substrates. The results reveal that the TNT signal detected on the unmodified substrate is relatively weak, whereas the substrate modified with capture molecules exhibits a significantly stronger TNT signal due to the adsorption of more TNT molecules. Supplementary Fig. 11s illustrates the performance of Mo₂C MXene-AuNBPs modified with $10^{-6}$ M and $10^{-5}$ M PATP in detecting 15 ppb TNT. According to the calculations of the capture molecule density in supporting information, each AuNBPs can connect with approximately 2.4 capture molecules. These capture molecules preferentially bind to the high-index surfaces[41], making capture molecules mainly localized in the hot spots of the AuNBPs and on the MXene surface.

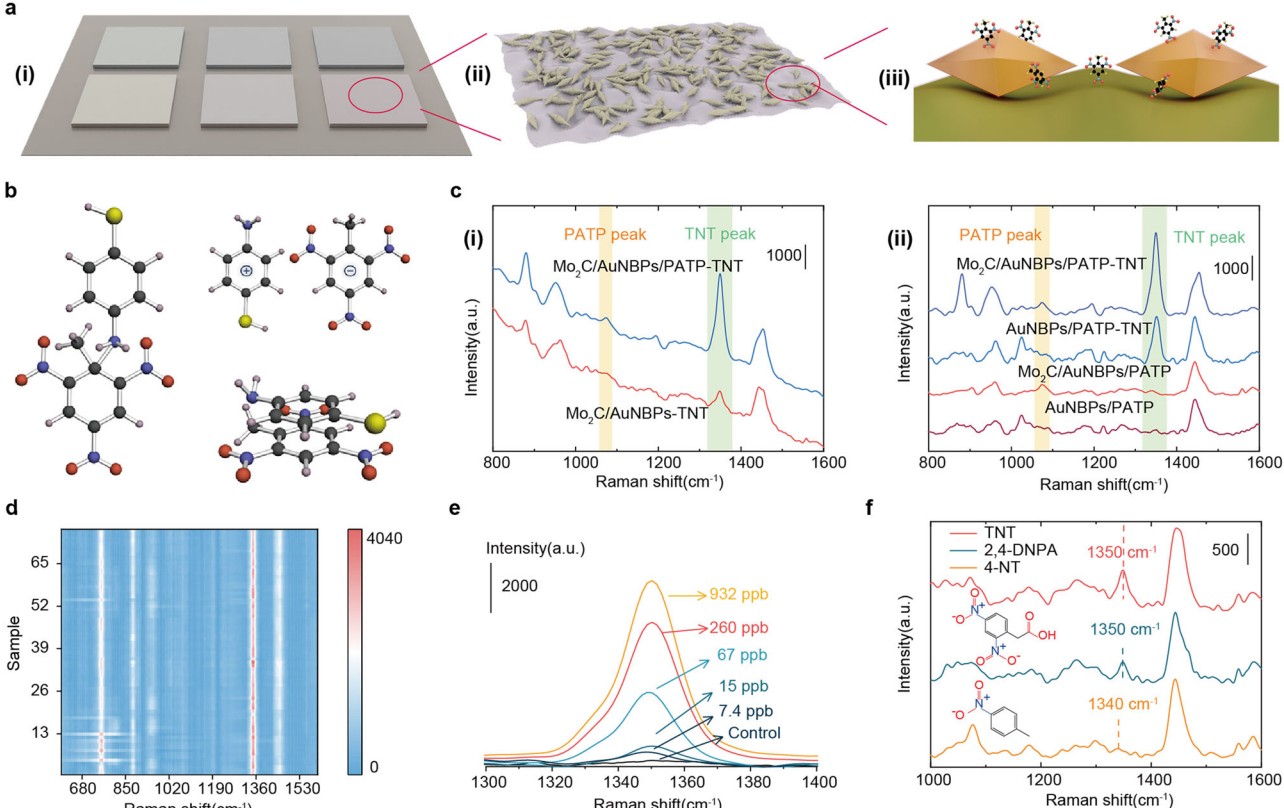

**Fig. 5 | SERS performance of Mo₂C MXene-AuNBPs. a** The method for detecting TNT and nitro compounds gas using Mo₂C MXene-AuNBPs. **b** Three interaction mechanisms between PATP and TNT. **c** (i) Raman spectra of 67ppb TNT detected by Mo₂C MXene-AuNBPs and Mo₂C MXene-AuNBPs-PATP. (ii) Raman Spectra of Mo₂C MXene/AuNBPs before and after modification with PATP for TNT detection. **d** Heatmap of Raman measurements at 75 random points along the coffee ring using Mo₂C MXene-AuNBPs-PATP to detect 67 ppb TNT. **e** Detailed Raman spectra of different concentrates of TNT in the range of 1300 cm⁻¹ to 1400 cm⁻¹ using Mo₂C MXene-AuNBPs-PATP. **f**. The Raman spectra of TNT, 4-NT, and 2,4-DNPA at 25 °C collected by Mo₂C MXene-AuNBPs-PATP.

In contrast, the modification with 40 μL of 10⁻⁵M capture molecules increased the density tenfold, resulting in a more dispersed distribution of PATP on the surface of the AuNBPs. This dispersion may cause the interacting TNT molecules to be located further away from the hot spots. Although a higher concentration of modified capture molecules may capture more TNT molecules, those located farther from the hot spots yield lower detected signals. Furthermore, the increased Raman signals from excess capture molecules may interfere with the detection of TNT signals, reducing the overall intensity of the detected TNT signals. The results in supplementary Fig. 11s indicate that the 10⁻⁶ M PATP-modified substrate achieves superior detection performance.

We also examine the relationship between adsorption time and the Raman signal of TNT detected by the substrate. Supplementary Fig. 12s (i) plots the Raman spectra obtained by detecting 15 ppb TNT molecules adsorbed onto the composite SERS substrate from 0 to 60 min. The quantification of the intensity of the TNT characteristic peaks is presented in Supplementary Fig. 12s (ii). It is evident that the intensity of the TNT peaks increases with adsorption time, but the rate of increase gradually diminishes. This suggests that the substrate is most efficient at capturing TNT molecules during the first 20 min, after which the adsorption efficiency of the composite SERS substrate gradually declines. To optimize detection efficiency and highlight the signal differences from adsorption, the study determined that the optimal concentration of the monolayer for modifying the SD-SERS arrays is 10⁻⁶ M, and the required detection time for nitro compounds is 20 min.

To further validate the superior SERS enhancement performance of Mo₂C MXene-AuNBPs, the AuNBPs were modified with capture molecules, and the same capture and detection conditions were applied to detect TNT. As shown in Fig. 5c (ii), the intensity of the TNT characteristic peak

at 1350 cm⁻¹ detected using Mo₂C MXene-AuNBPs-PATP and AuNBPs-PATP substrates was 1872.68 and 3349.62, respectively. This demonstrates that Mo₂C MXene-AuNBPs-PATP achieved better detection performance for TNT. Additionally, the larger specific surface area of MXene provides the substrate with stronger adsorption capacity, further improving the detection performance of Mo₂C MXene-AuNBPs-PATP for TNT. Next, the uniformity of the signal for TNT detection was evaluated. Figure 5d shows the results of Raman measurements at 75 random points along the coffee ring using Mo₂C MXene-AuNBPs-PATP to detect 67 ppb TNT. Supplementary Fig. 13s further plots the intensity of the TNT characteristic peak at 1350 cm⁻¹ across the 75 points, with the average value of 2993 and a relative standard deviation (RSD) of 9.82% which is affected by the differences in hotspot distribution at the testing points.

To validate the ability of the Mo₂C MXene-AuNBPs composite SERS substrate to detect TNT gas, the substrate was used to detect TNT at various concentrations (0.932 ppm, 0.260 ppm, 67.1 ppb, 15 ppb, and 7.4 ppb), the results are shown in Supplementary Fig. 14s. Figure 5e provides a detailed view of the Raman spectra of TNT in the range of 1300 cm⁻¹ to 1400 cm⁻¹, confirming that the prepared SERS substrate exhibits excellent detection performance. Furthermore, the detection capabilities were tested for two structurally similar compounds, 4-NT and 2,4-DNPA. Their intrinsic spectra are plotted in Supplementary Fig. 10s, with their main characteristic peaks appearing at 1340 cm⁻¹ and 1350 cm⁻¹, respectively. To evaluate the performance of the SERS substrate in detecting environmental chemicals, the experiments for TNT, 4-NT, and 2,4-DNPA detection were conducted under identical conditions, including the same pressure and temperature. Supplementary Fig. 15s shows the Raman spectra of TNT, 4-NT, and 2,4-DNPA detected at 60 °C using the Mo₂C MXene-AuNBPs-PATP substrate.

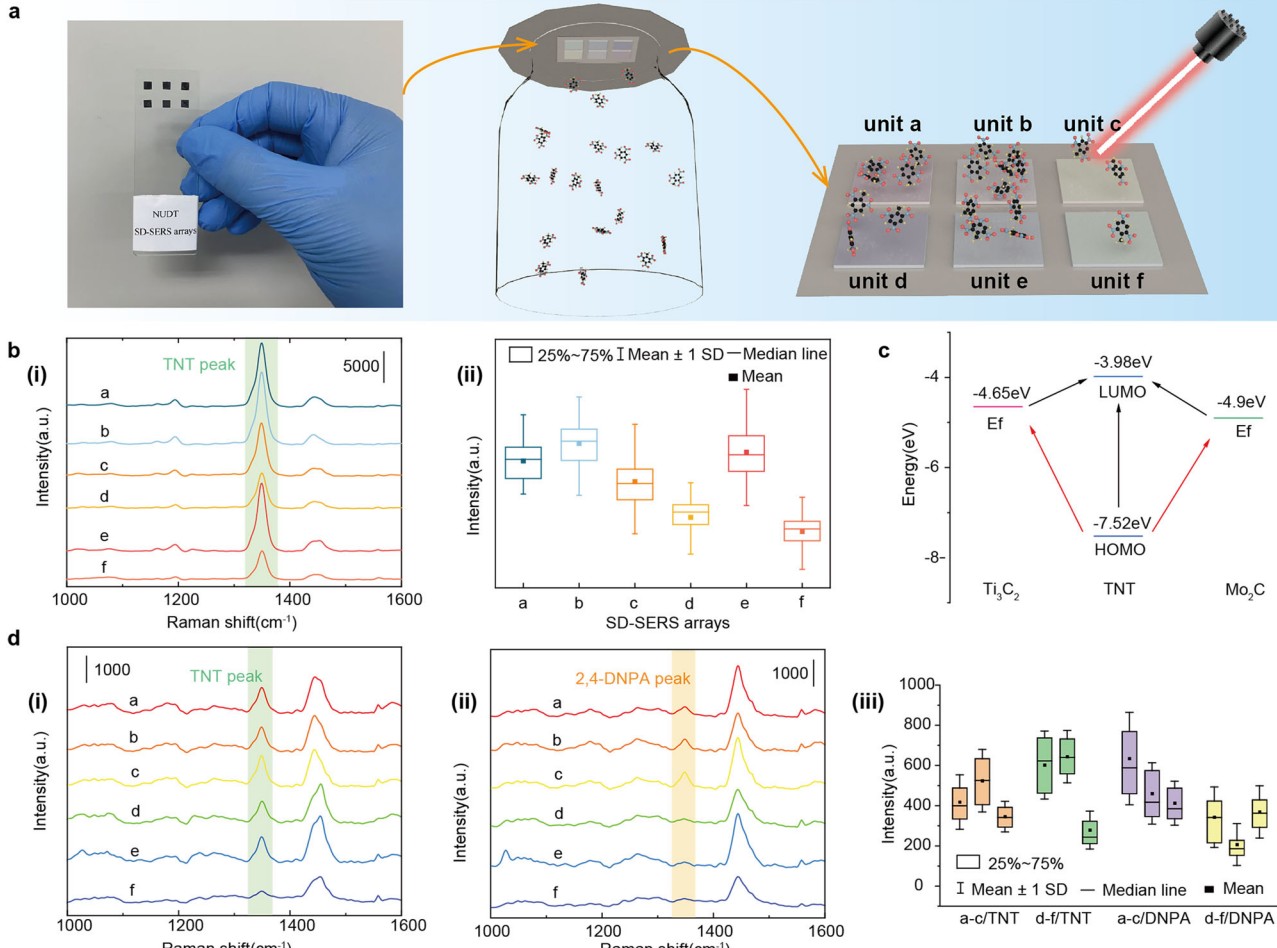

**Fig. 6 | SERS performance of SD-SERS arrays. a** The scheme for detection of test gases using SD-SERS arrays. **b** (i)The average Raman spectra and (ii) box plot for Raman intensity distributions at 1350 cm$^{-1}$ of detection for TNT gas at 60 °C using SD-SERS arrays. **c** The photo induced charge transfer between TNT[43] and two MXenes[28,42]. **d** (i) The average Raman spectra of detection for (i)TNT gas and (ii)2,4-

DNPA gas at 25 °C using SD-SERS arrays, and (iii) box plot for Raman intensity distributions at 1350 cm$^{-1}$ of detection for two kinds of gas at 25 °C using SD-SERS arrays. The vertical width of the box in box plots represents the interquartile range (IQR), which is the 25–75% range of the overall data arranged in descending order after excluding the influence of outliers.

It can be observed that the characteristic peaks of TNT and 2,4-DNPA appear at 1350 cm$^{-1}$, while the peak of 4-NT is at 1340 cm$^{-1}$, which are attributed to the symmetric stretching vibration of the NO$_2$. The intensity of the TNT characteristic peak is higher than those of 4-NT and 2,4-DNPA, demonstrating the high selectivity of the substrate in detecting TNT gases generated at high temperatures. However, when detecting the Raman spectra of TNT, 4-NT, and 2,4-DNPA gases at 25 °C, as shown in Fig. 5f, the nitro characteristic peaks of TNT and 2,4-DNPA remain observable, but the peak for 4-NT is no longer identifiable. Additionally, it was found that the Raman spectra of TNT and 2,4-DNPA gases at 25 °C in the range of 1000–1600 cm$^{-1}$ show significant overlap, making it impossible to directly distinguish the two substances based on their characteristic peak positions.

To distinguish between the highly overlapping spectra of TNT and 2,4-DNPA, SD-SERS arrays were employed to leverage differences in chemical enhancement and adsorption capacities, which is shown in Fig. 6a. This approach provides multidimensional intensity characteristics for different gas molecules. Figure 6b (i) shows the average Raman spectra of 180 measurements of TNT gas at 60 °C using six SERS substrate units (a-f) of the SD-SERS arrays. Units a-c correspond to Ti$_3$C$_2$ MXene-AuNBPs modified with PATP, 4MBA, and 6MNA, respectively, while units d-f correspond to Mo$_2$C MXene-AuNBPs modified with the same molecules. To ensure the collection of comprehensive data, the spectra were randomly sampled across the entire sensitive region of the substrates, including regions with non-uniform fabrication and coffee ring areas. This method increases the

reliability of the analysis by reflecting the true variability of the samples. The results demonstrate significant differences in the Raman intensity of the spectra obtained from the different SERS substrates, particularly in the intensity of the characteristic peak at 1350 cm$^{-1}$ when detecting TNT gas at 60 °C. Figure 6b (ii) further compares the characteristic peak intensities at 1350 cm$^{-1}$ for units a–f.

By comparing the spectral intensities of substrates based on the same MXene material but modified with different capture molecules, it was observed that Ti$_3$C$_2$ MXene-AuNBPs-PATP (unit a) exhibited slightly stronger spectral intensity than Mo$_2$C MXene-AuNBPs-PATP (unit d). According to the PICT principle (Fig. 6c), both MXene structures[28,42] can facilitate charge transfer with the LUMO orbitals of TNT[43] under 785 nm laser excitation. This demonstrates that both MXene-based substrates contribute to chemical enhancement, resulting in comparable detection performance. When analyzing the differences in signal intensities, a comparison of units a–c and d–f revealed that substrates modified with 6MNA (Ti$_3$C$_2$ MXene-AuNBPs-6MNA or Mo$_2$C MXene-AuNBPs-6MNA) displayed weaker characteristic peak intensities for TNT detection. This suggests that 6MNA interacts less effectively with TNT as a capture molecule. Consequently, SERS substrates with a 6MNA monolayer capture fewer TNT molecules, resulting in lower Raman signal intensities. To resolve the highly overlapping spectra of TNT and 2,4-DNPA, SD-SERS arrays were utilized to exploit differences in chemical enhancement and adsorption capacities. This approach

https://doi.org/10.1038/s42004-025-01656-2 **Article**

provided multidimensional intensity characteristics for distinct gas molecules.

To further validate the capability of SD-SERS arrays to distinguish between the highly overlapping spectra of TNT and 2,4-DNPA, Fig. 6d(i) and (ii) present the Raman spectra of TNT and 2,4-DNPA gases detected at 25 °C using SD-SERS arrays. Figure 6d(iii) quantifies the intensity of the characteristic peak at 1350 cm$^{-1}$ in the Raman spectra of both gases. For TNT, the chemical enhancement effects produced by the two MXene-based substrates are approximately similar. Consequently, TNT signals detected using the same MXene substrate modified with identical capture molecules exhibit minimal variation. However, substrates modified with 6MNA monolayers show notably weaker Raman signals, indicating reduced adsorption capability. These results align with the observations for TNT at 60 °C using SD-SERS arrays. In the case of 2,4-DNPA at 25 °C, the SD-SERS arrays demonstrate significant differences in chemical enhancement, with $Ti_3C_2$ MXene-AuNBP-based sensor units producing much stronger Raman signals compared to $Mo_2C$ MXene-AuNBP-based units. Further analysis of the PICT process for 2,4-DNPA (Supplementary Fig. 16s) shows that $Ti_3C_2$ MXene facilitates charge transfer with 2,4-DNPA, whereas $Mo_2C$ MXene is unable to achieve direct PICT under 785 nm laser excitation. This explains the observed differences in chemical enhancement for 2,4-DNPA when using SD-SERS arrays. Additionally, sensor units modified with PATP monolayers exhibit stronger Raman signals and superior adsorption capabilities. These findings demonstrate that SD-SERS arrays leverage differences in chemical enhancement and adsorption capabilities to provide multidimensional spectral features, enabling the differentiation of highly overlapping spectra for TNT and 2,4-DNPA, which is difficult to obtain from Raman spectra using a single substrate.

We subsequently evaluated the effectiveness of SD-SERS arrays in distinguishing TNT from the highly overlapping spectral structure of 2,4-DNPA. The random forest method, which incorporates machine learning (ML) algorithms, was employed for the classification of the two structurally similar compounds. During the data collection phase, for the detection of the same gas at the same temperature, 180 Raman spectral results were collected from each substrate as initial data for the machine learning model. The Raman spectral results from a total of six substrates, labeled a–f, under the same conditions, collectively constituted the original data set of the substrate array. To eliminate the impact of redundant data on model detection results, this study utilized the continuous numerical results of the characteristic peaks, specifically the Raman shifts in the range of 1300 cm$^{-1}$ to 1400 cm$^{-1}$, as the original feature vector of the substrates, as shown in Fig. 7a. For a single substrate, taking unit a as an example, the Raman intensity corresponding to the Raman shift from 1300 cm$^{-1}$ to 1400 cm$^{-1}$ in one detection was represented as the original feature vector for substrate a: $a = [a_1, a_2, \cdots, a_n]$, where n is the length of the characteristic peak values. The feature vector for the substrate array was expressed as $Nose = [N_{11}, N_{12}, \cdots, N_{6n}]$, resulting in a $6 \times n$ matrix, where 6 represents the number of individual substrates. As shown in Fig. 7b, to obtain the array feature vector, 180 mixed spectra were constructed by concatenating different batches of spectra selected from the data of unit a–f in the range of 1300 cm$^{-1}$ to 1400 cm$^{-1}$.

We also compared the capabilities of single unit a–f and SD-SERS arrays in distinguishing the two gases using Random Forests[44] (RF) based on confusion matrices by 10-fold cross-validation. Given the similarities in both position and intensity of the Raman characteristic peaks for the two gases at 25 °C, the primary focus was on the differentiation of the two gases under these conditions. As shown in Fig. 7c, the accuracy of the SD-SERS arrays model was significantly higher than that of the single detection unit model. Furthermore, to further analyze the obtained data results, Fig. 7d(i) and Supplementary Table 5s present the differences in accuracy between the single detection unit model and the SD-SERS arrays model. The accuracy of the single detection unit model was generally low. In contrast, the accuracy of the SD-SERS arrays model reached as high as 99.4%, demonstrating its optimal analytical capability. Thus, through the signal differentiation

approach of the single unit, the SD-SERS arrays achieved improved predictive accuracy for gases with similar structures by integrating differentiated spectral characteristics. Additionally, the precision, recall, and F1 score of the RF model were 99.3%, 99.4%, and 99.4%, respectively, confirming the effectiveness of the SD-SERS arrays.

To further compare the differences between various algorithms, we also evaluated the classification results of logistic regression[45], K-nearest neighbors (KNN)[46], and support vector machine (SVM) methods[47]. Figure 7d(ii) presents the classification results of the different methods for the prediction set. The conclusion drawn is that regardless of the method employed, the SD-SERS arrays consistently yielded the best results. Supplementary Table 6s lists the detailed numerical values of classification accuracy for the different methods. Recent studies have indicated that methods such as logistic regression, KNN, and SVM face challenges when extracting features from such high-dimensional data, leading to overfitting[48]. In this study, to avoid overfitting, we categorized the spectral intensity information solely at the characteristic peaks where the two similar substances were present, thereby reducing the feature complexity of the data. As a result, this study did not encounter overfitting issues with these methods. According to the results in Supplementary Table 5s, the classification accuracy using these methods with the SD-SERS arrays model reached 100%. Similar results were observed in the classification of two gases at 30 °C, as shown in Supplementary Figs. 17s, 18s. This demonstrates the superior algorithm adaptability of the SD-SERS arrays platform in the classification process of similar substances based on single characteristic peaks.

At the same time, we conducted a study on the concentration classification of TNT using the SD-SERS arrays model. Typically, the classification of substance concentrations often requires a clear linear relationship between the concentration of the substance and the intensity of the characteristic peaks. While machine learning classification methods can alleviate the need for linear relationships, errors resulting from detection or substrate preparation often affect the outcomes of concentration classification. For instance, taking the classification of TNT at 30 °C corresponding to concentrations of 15 ppb and 7.4 ppb at 27 °C as an example, Supplementary Fig. 19s show the classification results of different methods analyzing the data of TNT Raman peak in the 1300 cm$^{-1}$ to 1400 cm$^{-1}$ range with both the single substrate model and the SD-SERS arrays substrate model. From the confusion matrix for the RF method in Supplementary Fig. 19s as and the classification accuracy in Fig. 19s b(i), it can be observed that using the single substrate for classification of TNT characteristic peak data for concentrations of 15 ppb and 7.4 ppb yielded a maximum accuracy of 96.1%. Notably, for substrate unit e, this substrate model demonstrated good classification performance for similar substances but performed poorly in concentration classification. In contrast, the maximum accuracy for classifying TNT at concentrations of 15 ppb and 7.4 ppb using the SD-SERS arrays model reached 99.7%. Supplementary Table 6s and Fig. 19s b(ii) presents detailed results of the classification of the two TNT concentrations using different methods. This further confirms that the signal differentiation approach employed by the SD-SERS arrays platform not only yields favorable results in the classification of similar substances but also exhibits excellent potential in concentration classification. To further confirm the performance of SD-SERS arrays in classifying substances based on concentration, Fig. 8a presents the confusion matrix results of using a single unit and the SD-SERS arrays models with Random Forest (RF) for classifying different concentrations of TNT gases. It is evident that the concentration classification performance of the SD-SERS arrays model is more significantly outperform than that of the single unit model. Figure 8b shows the classification accuracy of different TNT concentrations using different models by RF, it is also clear that the SD-SERS arrays model exhibits excellent performance in concentration classification. The other ML results shown in Fig. 8c can also indicate that the SD-SERS arrays platform, utilizing a signal differentiation approach, not only demonstrates good efficacy in the classification of similar substances but also possesses remarkable potential in concentration classification.

**Fig. 7 | ML- assisted classification of TNT and 2,4-DNPA based on SD-SERS arrays. a**. Classification model based on Machine Learning methods. **b** Illustration of data processing for single unit and splicing spectra. **c** Results of distinguishing the two gases using Random Forest based on confusion matrices by 10-fold cross-validation. **d** (i) The RF classification performance of TNT and 2,4-DNPA gases at 25 °C using different models and (ii) the classification accuracy results of the different ML methods for the prediction set of TNT and 2,4-DNPA gases at 25 °C.

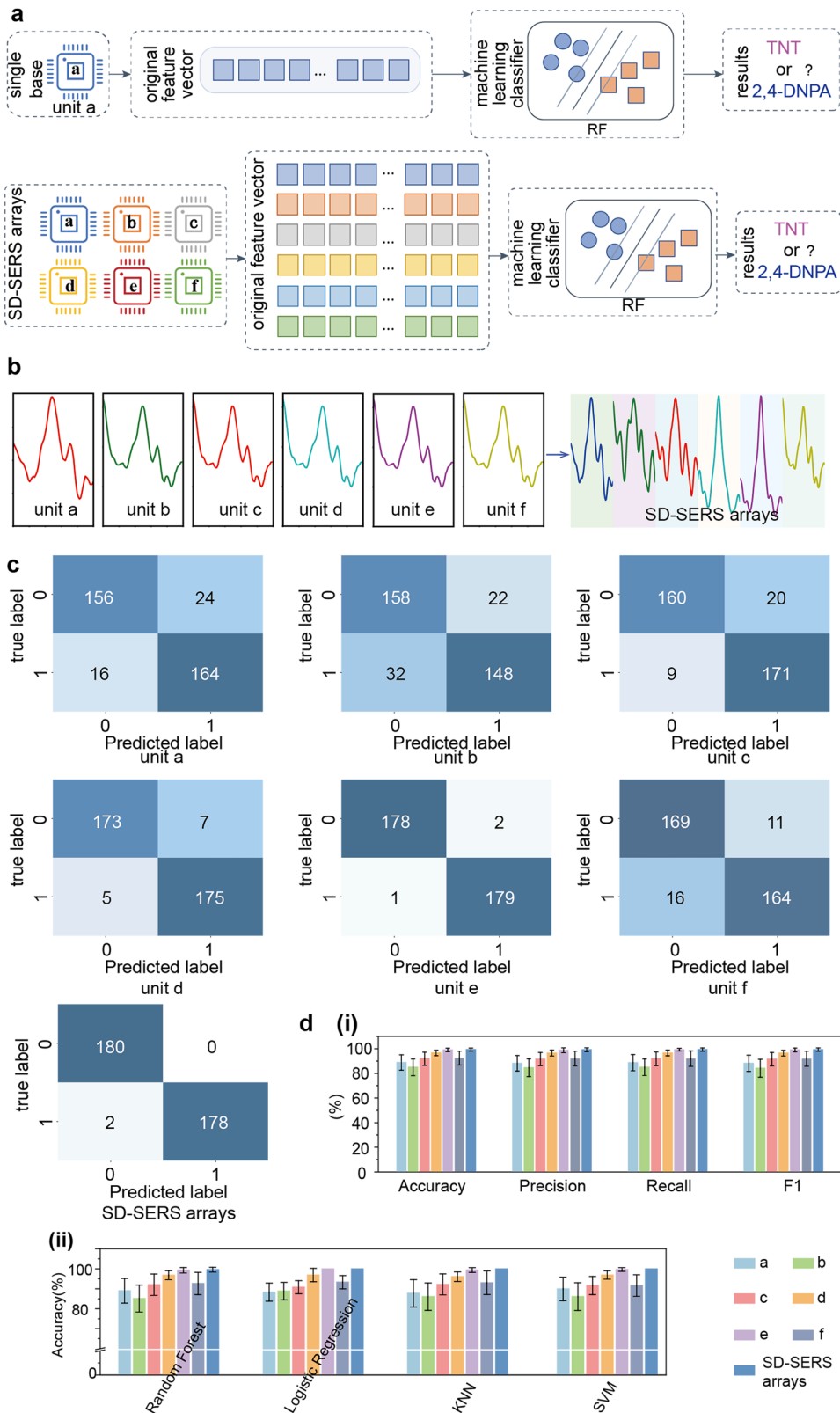

## Conclusion

In this study, we demonstrate the applicability of a signal differentiation-enabled SERS nose strategy for accurately distinguishing nitro compound gases. We successfully fabricated the MXene-AuNBPs-abs nanostructure by constructing a heterogeneous structure of SD-SERS arrays that integrates signal differentiation, electromagnetic enhancement, and adsorption capability differentiation. We then optimized the detection performance of various nanostructures using FDTD simulations, and further validated the performance of the signal differentiation and adsorption capability differentiation structures using first-principles calculations and experimental methods. Furthermore, we achieved trace detection of TNT gas using the integrated SD-SERS arrays composed of different units. Our research

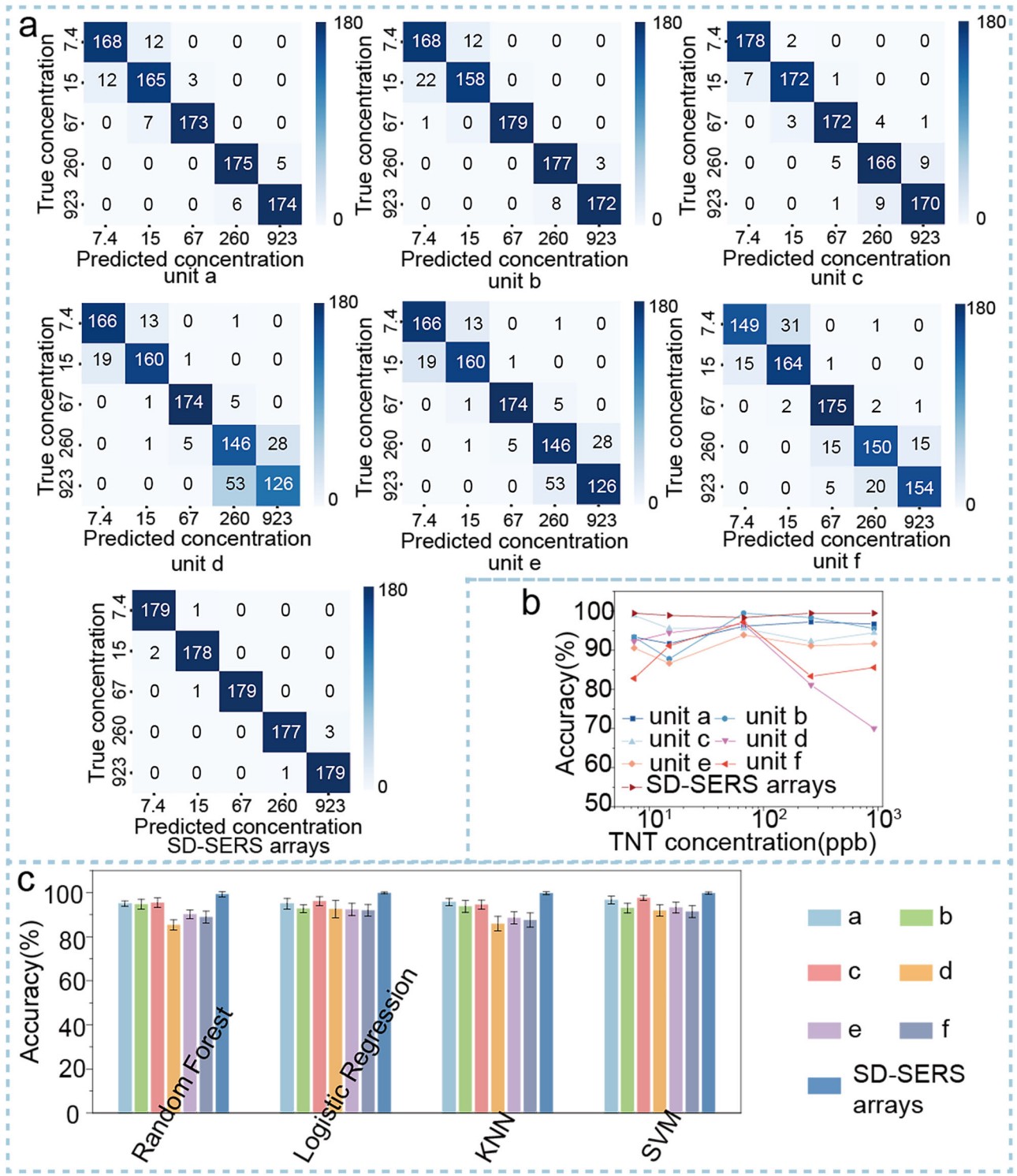

**Fig. 8 | ML- assisted classification of different concentrations of TNT based on SD-SERS arrays. a** The RF confusion matrix for the classification of different concentrations of TNT gases using different models. **b** The RF classification accuracy of different concentrations of TNT using different models. **c** The classification accuracy results of the different ML methods for the prediction set of different concentrations of TNT gases.

indicates that the information fusion from the SERS data provided by various MXenes and single molecular layer-constructed SD-SERS arrays offers a more comprehensive characterization of the samples. We then conducted detection of TNT and 2,4-DNPA gases, which exhibit highly overlapping spectra, and applied machine learning algorithms for classifying different gas types and varying concentrations of TNT gases. According to the experimental results, single substrates face limitations in sensitivity and selectivity when dealing with highly overlapping spectra. This makes it difficult to distinguish accurately between different substances or their concentrations in complex mixtures. In contrast, the SD-SERS array enhances differential recognition between similar compounds through its multidimensional data acquisition capabilities. In this study, the SD-SERS array demonstrated higher classification accuracy and stronger concentration prediction ability. This showcases that the SD-SERS arrays represent a convenient, effective, and highly accurate SERS nose technology. Given its versatility, this method could also be further applied to the

classification and identification of other samples, such as biological specimens. Due to the pathogenic nature of the analytes, we did not demonstrate the real-time detection capabilities of the SERS nose in an actual field deployment. However, all experimental procedures described in the paper can indeed be implemented in real-world environments. Therefore, the portability of the SERS nose allows it to be suitable for continuous sensing applications.

## Method

### Materials and instruments

2,4-Dinitrophenylacetic acid (2,4-DNPA), 4-nitrotoluene (4-NT), L-ascorbic acid (AA, 99.7%), Sodium hydroborate (NaBH$_4$, 99%), chloroauric acid hydrate (AuCl$_3$·HCl·4H$_2$O, 99.99%), cetyltrimethylammonium bromide (CTAB, 99%), hexadecyltrimethylammonium chloride (CTAC,99%), 4-Aminothiophenol (PATP, 99%), 4-Mercaptobenzoic acid (4MBA, 99%), 6-mercaptonicotinic acid (6MNA, 99%) were purchased from Sigma–Aldrich. 1-5 layers Ti$_3$C$_2$ MXene (1 mg/mL) and Mo$_2$C MXene (1 mg/mL) were from Jiangsu XFNANO Materials Co. Ltd. Hydrochloric acid (HCl, 36–38%) and tri-Sodium citrate (99.99%) were purchased from Sinopharm Chemical Reagent Co., Ltd. Silicon chips were bought from Academy of Aerospace Propulsion Technology. All the chemicals were used as received without purification. Ultrapure water (18.25 MΩ cm, 25 °C) was used in all experiments.

The morphologies of the nanostructures were obtained by scanning electron microscopy (SEM, Zeiss Sigma 300). The morphologies of nanostructures were characterized by transmission electron microscopy (TEM; FEI Talos F200S). Zeta potential analysis was conducted using the Malvern Zetasizer Nano ZS90. The elemental compositions of SERS substrates were analyzed by energy-dispersive spectroscopy (GatanGIF Quantum 963). UV−vis was analyzed by a UV probe (UV-2600, Shimadzu). The Raman spectra were measured by a Raman microscope (BWS415-785S, B&W, Tek, Newark, DE) excitation laser.

### Synthesis of gold nanobipyramid

The seed-mediated growth method for synthesizing AuNBPs primarily enhances the twin crystal quality of the seeds[49,50]. This process is divided into two main steps:

Synthesis of AuNBPs Seeds: 5 mL of 0.1 M CTAC solution is added to a 50 mL flask, which is stirred vigorously at 30 °C. Subsequently, 0.5 mL of 5 mM tetrahydroauric acid aqueous solution, 0.5 mL of 0.1 M sodium citrate solution, and 4.5 mL of deionized water are sequentially added to the flask. Once the solution is well mixed, 250 μL of 25 mM fresh sodium borohydride solution is added, and the mixture is stirred vigorously for 2 minutes at 30 °C. After sealing the flask, it is stirred at 80 °C for 240 min. Upon completion of the reaction, the mixture is stored at 4 °C for later use.

Preparation of AuNBPs: 10 mL of 0.1 M CATB is added to a 50 mL reaction flask, and while stirring vigorously at 30 °C, 500 μL of 10 mM tetrahydroauric acid solution, 100 μL of 10 mM silver nitrate, and 200 μL of 1 M HCl are sequentially introduced. The mixture is stirred until homogeneous, followed by the addition of 80 μL of 0.1 M ascorbic acid solution, resulting in a color change from light yellow to colorless. Afterward, 70 μL of the previously synthesized gold nanodumbell seeds is immediately added to the growth solution. After stirring vigorously for 10 min, the stirrer is removed, and the solution is allowed to stand at 30 °C for 2 h. At the end of the reaction, the product is centrifuged at 9500 rpm, washed to remove the supernatant, and the final product is resuspended in 4 mL of deionized water.

### Fabrications of MXene-AuNBPs

The preparation of MXene-AuNBPs composite structures is primarily conducted through self-assembly method assisted by electrostatic attraction[31,51]. Firstly, Ti$_3$C$_2$ MXene-Mo$_2$C MXene were sonicated in an ultrasonic bath continuously for 2 h below 25 °C to disperse MXene. 200 μL of MXene was mixed with 2 mL of AuNBPs solution. After ultrasonic

treatment for 2 h at room temperature, the resulting MXene-AuNBPs composite product was centrifuged with 10,000 rpm for 10 min to remove unconnected AuNBPs, and the product was then resuspended in 600 μL of deionized water.

### Preparation of single-molecule-layer modified MXene-AuNBPs

Add 40 μL of TNT capturers (PATP, 4MBA, and 6MNA) at different concentrations ($10^{-6}$ M) into the prepared MXene-AuNBPs solution. Incubate the mixture under agitation for 1 h. After incubation, centrifuge the sample at 8000 rpm for 10 minutes. Remove the supernatant and resuspend the precipitate in 600 μL of deionized water.

### Fabrication of SD-SERS arrays

After modifying two types of SERS substrates (Ti$_3$C$_2$ MXene and Mo$_2$C MXene) with three different capturers (PATP, 4MBA, and 6MNA), six SERS substrates were obtained, each in a volume of 600 μL, and named as follows: Ti$_3$C$_2$ MXene-AuNBPs-PATP、Ti$_3$C$_2$ MXene-AuNBPs-4MBA、Ti$_3$C$_2$ MXene-AuNBPs-6MNA、Mo$_2$C MXene-AuNBPs-PATP、Mo$_2$C MXene-AuNBPs-4MBA, and Mo$_2$C MXene-AuNBPs-6MNA, corresponding to sensor a, b, c, d, e, and f, respectively.

Take 5 μL each of sensor a–f, and drop them separately onto the surface of six 2 mm × 2 mm silicon wafers. Place the sensors on a heating plate at 70 °C for annealing. After drying, add another 5 μL of substrates a–f to their respective silicon wafers, followed by annealing. Repeat this process three times to obtain sensors a to f. These sensor arrays collectively form the SD-SERS arrays.

### SD-SERS arrays for detecting gas molecules

Place powdered samples of different chemical molecules (TNT, 2,4-DNPA, and 4-NT) into capped glass bottles. Heat the bottles at different temperatures for 48 hours to generate gas molecules at varying concentrations. Attach the prepared SD-SERS arrays to a Petri dish using high-temperature conductive adhesive. Open the bottle cap and position the Petri dish over the bottle's opening, ensuring the side with the SD-SERS arrays is exposed to the chemical atmosphere for adsorption of the gas molecules. After a set exposure time, remove the Petri dish and perform Raman spectroscopy analysis. The SD-SERS arrays were placed under a Raman microscope, and Raman spectral information was collected using a BWS465-785H Raman laser with a 785 nm excitation laser with an integration time of 10 s and power of 30 mW.

### Machine learning-assisted classification and detection

For unit a–f of SD-SERS arrays, when detecting TNT, 4-NT, and 2,4-DNPA gases under varying temperatures and durations, the characteristic peak signals were modeled into a feature matrix representing the SERS nasal sensor. To train our machine learning model, we conducted gas detection at different temperatures. We selected 60 random points on each individual substrate of the SD-SERS arrays for detection and then prepared new SD-SERS arrays for further testing. This process was repeated three times, resulting in 180 spectral data points from each individual substrate. Overall, we collected a total of 16,200 signal data points from the SD-SERS arrays across various gases and concentrations. Specifically, a SD-SERS arrays detection sample can be characterized as a 6×N matrix, where 6 represents unit a-f, and N denotes the length of the signal values within the characteristic peak range. During the analysis of machine learning classification performance, TNT, 4-NT, and 2,4-DNPA gases were assigned classification labels of 0, 1, and 2, respectively. Each dataset entry consists of a SD-SERS arrays feature matrix and its corresponding classification label. In this study, a total of 16,200 data entries were collected. For temperature and duration variables, the dataset was split into training and testing sets in an 8:2 ratio. The machine learning model was trained on the training set to learn the relationship between the SERS nasal sensor samples and their labels. The model then predicted labels for the unlabeled testing set to

validate classification performance. To ensure experimental rigor and reliability, the experimental results were averaged over 10 random splits of the dataset. This study employed four commonly used evaluation metrics in machine learning classification: precision, recall, accuracy, and F1.

## Data availability

The original data displayed in the main manuscript are provided in Supplementary Data 1. Dataset of Supplementary Information is provided in Supplementary Data 2. Any other relevant data are available from the corresponding author upon reasonable request.

## Code availability

The source code is available in a public repository: https://github.com/trwang92/SERS_nose.

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

## Acknowledgements

P.D. acknowledges the National Natural Science Foundation of China (No.51475468). T.W. acknowledges that his primary affiliation is Wenzhou University of Technology. The authors acknowledge National Supercomputing Center in Shenzhen Computing Center for the technique support in simulations.

## Author contributions

T.W. provided the original idea. P.D. and X.W. jointly supervised the project. T.W. and P.D. carried out the design of the experiment, prepared the measurements, and wrote the manuscript. H.Y. and L.K. completed the machine learning algorithm. W.T. and W.Q. participated in the DFT and FEM simulation calculations. D.X., X.W., S.X., X.C., L.Y., and Q.F. participated in the characterizations and discussions. T.W., P.D., and W.Q. corrected the manuscript. All authors have approved the final manuscript.

## Competing interests

The authors declare no competing interests.
