## [Transparent Peer Review file · Communications Chemistry]

SERS nose arrays based on a signal differentiation approach for TNT gas detection

Corresponding Author: Dr Tianran Wang

Version 0:

Reviewer comments:

Reviewer #1

(Remarks to the Author)

The manuscript submitted by Dong et al. describes gold nanoparticle and MXene heterostructures for detection of TNT using the materials interface for signal differentiation in surface enhanced Raman scattering (SD-SERS). The authors hypothesize that charge transfer can lead to distinguishable spectral signals. The data in the paper appears rigorous but the organization of the figure and texts makes the manuscript difficult to follow. Each figure has many panels and some are very small making it difficult to see the data, even in full screen mode. For example, Figure 2d the lines are so faint it is not easy to read the absorption maxima positions. The figure captions do not thoroughly explain the data. This should be improved throughout the manuscript. The corresponding explanation in the text could be a few pages away so it is tedious to go back and forth between text and data to understand the experimental details. If no figure limitation, it would be a big improvement to have more figures with less panels are readable and the text can flow with the figures more seamlessly. I list specific comments and questions about the manuscript below.

On line 59-62, a discussion of a study by Yang et al. mentions detection at levels of 100 ng. Was this performed in liquid or vapor phase? Are the units ng/L?

Figure 1 lists 6 SERS substrates based on a combination of 2 different MXene surfaces and 3 different chemical functionalities. The data in Figure 4 is hard to determine if there is significant differences. It is not clear what the color coding is in g (ii). Are there trends that follow adsorption energy? The data in the box varies by 25 – 75% of the mean? The description of how they defined the vertical width of the boxes in the box plot is unclear to me.

On line 194, the authors state the particles exhibit minimal aggregation and later state the AuNBPs have a closely packed arrangement. From the TEM images, there does appear to be multilayers in some places. Thus it appears there is aggregation. SEM images over a larger area would provide a better assessment of aggregation over large area.

In regards to Fig. 2i and h, the authors state on line 197 that there is successful connection between MXene and AuNBPs. I am not sure what this means. From the data, the conclusion seems to simply be that they have deposited the particles on the surface.

Figure 3 is better to discuss the interactions between AuNBPs and the surface. Fig. 3c the text under the figures is blurry so the adsorption energies would be better in the caption. Was the adsorption energy for 3MNA calculated? In the same figure the spectral data is hard to see with the text over the data. Thus trends they are trying to convey are unclear. It is interesting that the case of Ti3C2 with 4MBA has a much different adsorption energy than the others. Was there noticeable changes in SERS signal of SERS data due to this energy difference? I also don't understand why they chose to plot 4MBA and PATP in e but only PATP in f. It is hard to determine what point they are trying to make. They discuss the difference in SERS signal on lines 255-57 but I am wondering if the difference in adsorption energies on Ti3C2 leads to different charge transfer and SERS signal.

Why was the PATP functionalized surface chosen for detection of TNT?

On line 337 the authors mention the RSD was 9.82%. What is the average value of the signal? Is the signal with and without Au NBPs correspond to the magnitude of the field enhancement E^4 in the hotspots due to configurations shown in Fig. 3a. Is

that the source of RSD?

Line 347, "Raman spectra of TNT, 4-NT, and 2,4-DNPA detected at 60°C using the Mo₂C MXene-AuNBP-PATP substrate"
Why was Mo₂C chosen over Ti₂C₂ that had higher signal in Fig. 3?

Line 228 has a typo, maybe a missing reference at the end.

Figure 5 is very busy and hard to read. The values for accuracy, precision are hard to differentiate on the very small figure as well as the comparison of the ML methods, for example.

Reviewer #2

(Remarks to the Author)

Referee report: SERS nose arrays based on a signal differentiation approach for TNT gas detection

MS No: COMMSCHEM-25-0306-T

Decision: Major revision

1. The authors highlight the SERS array as a key advantage, but it is unclear whether the full array is needed for TNT detection or if a single element suffices. This important detail is missing from the abstract and conclusion and should be clarified for better understanding of the sensor's practical application.
2. PTAP, 4MBA, or similar functional molecules are already adsorbed onto a metal surface via physical interactions, what percentage of their functional groups or active sites remains available for TNT detection? Have the authors quantified or estimated the degree of surface accessibility post-adsorption, and how does this affect the sensitivity and reliability of the detection mechanism?
3. Although the metal substrate and the laser excitation wavelength (785 nm) are in resonance, TNT itself does not exhibit significant electronic absorption near this wavelength. Since the interaction with TNT is mediated via the attached analyte molecule, could the authors clarify how effectively TNT molecules are localized within the plasmonic 'hot spots' of the metal nanostructures? Additionally, it would be helpful to understand whether the spatial distribution and availability of TNT in these hot spots have been quantified or considered in evaluating the sensitivity and reproducibility of the detection.
4. The term 'physical coupling methods' is not standard or commonly used in the context of nanoparticle synthesis, particularly for Ag and Au nanoparticles. It would be helpful if the authors could clarify what they mean by this term or consider using more widely accepted terminology such as 'physical synthesis methods' or specify the particular techniques involved (e.g., laser ablation, evaporation-condensation). Clear definitions will improve the clarity and scientific rigor of the manuscript.
5. Although the authors mention control experiments for selectivity, the discussion and experimental details provided are unclear and somewhat confusing. The manuscript lacks a systematic and well-structured presentation of control data that convincingly demonstrates the sensor's ability to distinguish TNT from structurally similar compounds or potential interferents. For example, competitive binding experiments involving TNT and related analytes would provide stronger evidence of selectivity. Before performing comparisons with machine learning data obtained from the electronic nose, it is important to first establish a clear and robust baseline through well-designed control and selectivity experiments.
6. The authors describe an approach where the SERS substrate is incubated with gas-phase analytes and then removed for Raman measurement. While this method may be effective for controlled lab studies, it raises questions about its applicability for real-time or in situ detection. The manuscript would benefit from a discussion on whether the setup can be adapted for continuous or portable SERS-based sensing, as real-time detection is crucial for practical, field-deployable applications.

Further comments below require additional attention:

7. Line 37: Different style of citation. Please check. It even starts with citation 2, afterwards style is changed. Is the citation [2] the same as 2?
8. Line 58: Check misspelling, such as 'senors' instead of sensors.
9. Line 81: The example of food safety needs citation(s).
10. Line 93-95: Why is electromagnetic enhancement no key factor?
11. Line 105-107: What is a physical enhancement? Please clarify the role of EM enhancement.
12. Line 112/113: "the physical enhancement structure of the SD-SERS arrays should exhibit superior electromagnetic enhancement hotspots". Why 'should'? The arrays exhibit EM hotspots!
13. Line 114: "physical enhancement "hotspots". Better EM enhancement than physical enhancement.
14. Line 118: "with a tip-to-tip distance of 1.5 nm in all three cases". Is this a realistic value for fabrication of these structures?
15. Line 131: "[...] and chemical enhancement properties were obtained". Do they show no EM enhancement?
16. Line 161: CTAB instead of CTAC?
17. Line 171/172: "This resonance enhances the intrinsic SERS capability of the substrate, thereby improving its overall performance." Misleading, please improve the sentence.
18. Line 229/230: Are the "head-to-head, head-to-shoulder, and shoulder-to-shoulder configurations" realistic for the fabricated SERS substrates?
19. Line 234: beginning of the line. Is there a reference missing?
20. Line 249: What about stronger EM enhancement in those cases the analyte is closer to the surface due to attractive interaction?
21. Line 288 sentence is incomplete "According to (), MXene also have smaller SERS performance than Ti₃C₂"
22. In Fig. 3e and 3f, two different styles to highlight the PATP modes were used and in one case 2 modes are marked and

in the other only 1 mode. Why?

23. Fig. 3e and 3f is lacking from visibility. It is too small. Spectral features are barely visible. Please improve.

24. Fig. 4: To which vibration the peak around 1440 cm⁻¹ is assigned?

25. Line 301: How the formation of a monolayer is confirmed?

26. Line 319/320: Is a change of the signal observed for PATP after interaction with TNT and is this useable for a concentration-dependent measurement of TNT or similar molecules? Or is the analysis only based on the specific TNT Raman mode?

27. Line 346: The concentration values of TNT need to be related with relevant concentrations in real application scenarios.

28. Line 347/348: Please mention here the band assignment.

29. Line 389-391: How to deal with the significant differences in Raman intensity across the various SERS substrates? How to define the measurement procedure that stable and comparable measurement conditions are guaranteed?

30. Line 428: How many spectra per concentration? How many replicates? It should be at least 3 replicates tested on 3 different SERS substrates per concentration value for statistical relevance.

31. Line 476: "[...] but performed poorly in concentration classification". Does that mean that a quantification is not achieved? What is the required detection sensitivity for the target analytes? And is this concentration value identified?

Reviewer #3

(Remarks to the Author)

This peer review report was co-authored as part of the Communications Chemistry initiative to support the training and recognition of Early Career Researchers in the peer review process.

Version 1:

Reviewer comments:

Reviewer #1

(Remarks to the Author)

The authors have addressed my concerns in the first submission. The manuscript reads more clearly. The benefit of using the SD-SERS arrays is much clearer as a result of the reorganization of the manuscript.

Reviewer #3

(Remarks to the Author)

The authors have adequately addressed all the comments raised in our previous review. The current version is acceptable and can be recommended for acceptance, pending minor editorial suggestions or formatting adjustments as per the journal's requirements.

Reviewer #1 (Remarks to the Author):

The manuscript submitted by Dong et al. describes gold nanoparticle and MXene heterostructures for detection of TNT using the materials interface for signal differentiation in surface enhanced Raman scattering (SD-SERS). The authors hypothesize that charge transfer can lead to distinguishable spectral signals. The data in the paper appears rigorous but the organization of the figure and texts makes the manuscript difficult to follow. Each figure has many panels and some are very small making it difficult to see the data, even in full screen mode. For example, Figure 2d the lines are so faint it is not easy to read the absorption maxima positions. The figure captions do not thoroughly explain the data. This should be improved throughout the manuscript. The corresponding explanation in the text could be a few pages away so it is tedious to go back and forth between text and data to understand the experimental details. If no figure limitation, it would be a big improvement to have more figures with less panels are readable and the text can flow with the figures more seamlessly. I list specific comments and questions about the manuscript below.

Response: We thank the reviewer for their suggestions regarding the figures and text in our paper. We take the reviewer's concern about the complexity of the design, which makes it difficult to read, very seriously. In response, we have redesigned and reorganized the images in the results section. Additionally, we have simplified the language in the discussion section. The following are the modifications made to the image section.

1. We have reorganized Figure 2. The original Fig 2d has been moved to Supporting Information Figure 5, allowing readers to more easily observe the effects of different gold seeds on the morphology of gold nanodimers and their absorption peaks in the supporting materials.
2. Additionally, we have enlarged the original images in Fig 2c, 2k, and 2l to enable readers to more clearly identify the information presented in those figures. The new Fig 2 we redrawn is shown below.

And we have streamlined the text in the corresponding paragraph, such as the original statements are changed from line 179-185 "The surface potentials of Ti_3C_2 MXene and Mo_2C MXenes, as well as the surface potential of AuNBPs, were characterized. For

Ti₃C₂ MXene, the surface potential is -35.8 ± 0.87 mV, with detailed data shown in Supplementary Fig. 5s(a). Supplementary Table 1s provides the peak and area information for the zeta potential of Ti₃C₂ MXene. These results confirm that both MXene surfaces carry negative charges due to the presence of -OH and -COOH functional groups. As shown in Fig. 2k, the surface potential of Mo₂C MXene is -26.1 ± 0.83 mV. Detailed surface potential information for Mo₂C MXene is provided in Supplementary Fig. 5s(b), and Supplementary Table 2s presents the peak and area information for the zeta potential of Mo₂C MXene.” to “The surface potentials of Ti₃C₂ MXene and Mo₂C MXenes, as well as the surface potential of AuNBPs, were characterized at Fig 2d. For Ti₃C₂ MXene and Mo₂C MXene, the surface potential is -35.8 ± 0.87 mV with detailed data shown in Supplementary Fig. 6s(a) and -26.1 ± 0.83 mV with detailed data shown in Supplementary Fig. 6s(b), respectively. Supplementary Table 1s and Table 2s provides the peak and area information for the zeta potential of Ti₃C₂ MXene and Mo₂C MXene. These results confirm that both MXene surfaces carry negative charges due to the presence of -OH and -COOH functional groups.”

The new Figure 2

3. We divided the original Figure 3 into two parts. The new Figure 3 focuses on the coupling hotspot effect between AuNBPs and the electromagnetic field enhancement in MXene-AuNBPs. The new Figure 4 describes the charge transfer behavior between different MXene materials and adsorbed molecules, and experimentally validates that the differences in adsorption behavior can lead to variations in the intensity of the enhanced Raman signals from different MXene-AuNBPs. The new figures are shown as follow. The corresponding statements have also been modified.

The new Fig 3

The new Fig 4

4. To further enhance the readability of the paper, we have divided the original Figure 4 into new Fig 5 and Fig 6, which are shown as follow. The new Figure 5 mainly presents the SERS performance of a single substrate, Mo₂C MXene-AuNBPs, in detecting TNT and other nitro compounds in gaseous form. The new Figure 6 analyzes the signal differences when detecting TNT and 2,4-DNPA in various environments using six different substrates (a-f) to form SD-SERS arrays.

The new Fig 5

The new Fig 6

Additionally, these new images were placed alongside the corresponding text to enhance the article's readability and reduce the frequency with which readers have to flip pages to find relevant images.

- we have also removed redundant sections in this part, such as deleting "Fig. 4g(i) shows the average Raman spectra from 180 measurements of TNT gas at 60°C using six SERS substrate units (a-f) of the SD-SERS arrays. Units a-c correspond to Ti₃C₂ MXene-AuNBPs modified with PATP, 4MBA, and 6MNA, respectively, while units d-f correspond to Mo₂C MXene-AuNBPs modified with the same molecules.", which is the section that repeats with "Figure 4g (i) shows the average Raman spectra of 180 measurements of TNT gas at 60°C using six SERS substrate units (a-f) of the SD-SERS arrays. Units a-c correspond to Ti₃C₂ MXene-AuNBPs modified with PATP, 4MBA, and 6MNA, respectively, while units d-f correspond to Mo₂C MXene-AuNBPs modified with the same molecules." and so on.
- The original Figure 5 made it difficult for readers to understand the layout. To address this, we rearranged the positions of the sub-images in the figure. Additionally, we

categorized each section's images using dashed boxes and adjusted the font size to enhance the image clarity. These changes enhance the readability of the figures in the paper. The modified figure is now referred to two parts: Part one is the new Figure 7 which discusses the ML- assisted classification of TNT and 2,4-DNPA based on SD-SERS arrays and the new. Part two is the new Fig 8 which discusses ML- assisted classification of different concentrations of TNT based on SD-SERS arrays. The two new figures are show as follow.

The new Fig 7| **ML- assisted classification of TNT and 2,4-DNPA based on SD-SERS arrays.** a. Classification model based on Machine Learning methods. b. Illustration of data processing for single unit and splicing spectra. c. Results of distinguishing the two gases using Random Forest based on confusion matrices by 10-fold cross-validation. d. (i) The RF classification performance of TNT and 2,4-DNPA gases at 25°C using different models and (ii) the classification accuracy results of the different ML methods for the prediction set of TNT and 2,4-DNPA gases at 25°C.

The new Fig 8| **ML- assisted classification of different concentrations of TNT based on SD-SERS arrays.** a. The RF confusion matrix for the classification of different concentrations of TNT gases using different models. b. The RF classification accuracy of different concentrations of TNT using different models. c. The classification accuracy results of the different ML methods for the prediction set of different concentrations of TNT gases.

On line 59-62, a discussion of a study by Yang et al. mentions detection at levels of 100 ng. Was this performed in liquid or vapor phase? Are the units ng/L?

Response: We appreciate the reviewer's attention to this study. The paper states that " the detection limits toward three representative nitro-explosives, namely trinitrotoluene, pentaerythritol tetranitrate, and cyclotetramethylene tetranitroamine, are as low as 500, 100, and 50 ng, respectively." The corresponding detection limit information has been directly restated in this paper.

Actually, the researchers used an electronic nose (E-nose) to detect gas products generated by the photodecomposition of explosives under UV light, rather than directly detecting the explosive molecules or their saturated vapor. Specifically, the method involved placing varying amounts of explosive powder in a photodecomposition chamber,

where the sensor identified the presence of explosives by detecting the nitrogen dioxide gas produced after photolysis. Therefore, the reported detection limits for the explosives refer to mass units, not solution or gas concentration units.

Figure 1 lists 6 SERS substrates based on a combination of 2 different MXene surfaces and 3 different chemical functionalities. The data in Figure 4 is hard to determine if there is significant differences. It is not clear what the color coding is in g (ii). Are there trends that follow adsorption energy? The data in the box varies by 25 – 75% of the mean? The description of how they defined the vertical width of the boxes in the box plot is unclear to me.

Response: Thank you for your' suggestions regarding the figures. We have reformatted the images for improved clarity, and the new Figure 6 is displayed below. To confirm the ability of SD-SERS to differentiate data, the box plots in the original Figure 4(the new Fig 6) show the characteristic peaks for each unit in the SD-SERS arrays when detecting gases individually. It is evident that the signal intensities vary across different units in the new Fig 6b(ii). Although the intensity differences for TNT detection at low concentrations in some units (original Figure 4i(i), now Figure 6c(i)) are not significant, such as units a, b, d, and e are similar during TNT detection. In contrast, there are clear differences in signal intensities among the different units when detecting 2,4-DNPA. The differentiated signal behaviors generated by these SD-SERS arrays when detecting various substances provides a solid basis for distinguishing between the two gases with highly overlapping spectra.

The two box plots in the original Figure 4 (Fig 4g(ii) and 4i(iii)) are statistical charts used to display data distribution. They illustrate the statistical characteristics of a data set, including the lower limit, first quartile (Q1), median, third quartile (Q3), and upper limit. The bottom and top of the box represent the first and third quartiles, respectively. In statistics, quartiles are values that divide the data into four equal parts when arranged in ascending order. Each part contains 25% of the data, and the values at the quartile points are referred to as quartiles. There are three quartiles: the first quartile is the lower quartile, the second quartile is the median, and the third quartile is the upper quartile. Therefore, the range of data variation is not simply defined as 25%-75% of the mean. Instead, it refers to the 25%-75% percentile range of the overall data arranged in descending order after excluding the influence of outliers. To facilitate readers' understanding of the box plot, we have included

a description of the box plot in the figure legend as “The vertical width of the box in box plots represents the interquartile range (IQR), which is the 25%-75% range of the overall data arranged in descending order after excluding the influence of outliers.”.

The new Fig 6

On line 194, the authors state the particles exhibit minimal aggregation and later state the AuNBPs have a closely packed arrangement. From the TEM images, there does appear to be multilayers in some places. Thus it appears there is aggregation. SEM images over a larger area would provide a better assessment of aggregation over large area.

Response: We thank the reviewers for their suggestions. The SEM images have been added as Figure 7s e in the supplementary information.

In regards to Fig. 2i and h, the authors state on line 197 that there is successful connection between MXene and AuNBPs. I am not sure what this means. From the data, the conclusion seems to simply be that they have deposited the particles on the surface.

Response: We thank the reviewer for the question regarding the connection between MXene and AuNBPs. In the synthesis of MXene-AuNBPs, the connection is primarily achieved through electrostatic adsorption, based on the surface charges of MXene and AuNBPs. To enhance clarity, the content in Figure 2 has been reorganized. The mechanism

of the connection between MXene and AuNBPs has been reformulated and presented as new Figures 2d and 2e. Additionally, we have rephrased this section to provide clearer information.

Figure 3 is better to discuss the interactions between AuNBPs and the surface. Fig. 3c the text under the figures is blurry so the adsorption energies would be better in the caption. Was the adsorption energy for 3MNA calculated? In the same figure the spectral data is hard to see with the text over the data. Thus trends they are trying to convey are unclear. It is interesting that the case of Ti3C2 with 4MBA has a much different adsorption energy than the others. Was there noticeable changes in SERS signal of SERS data due to this energy difference? I also don't understand why they chose to plot 4MBA and PATP in e but only PATP in f. It is hard to determine what point they are trying to make. They discuss the difference in SERS signal on lines 255-57 but I am wondering if the difference in adsorption energies on Ti3C2 leads to different charge transfer and SERS signal.

Response: Thank you for the reviewer's suggestions. We have reorganized the original Figure 3, and the new Figure 3 now focuses solely on the interactions between AuNBPs. The new Figure 4 illustrates the signal differentiation approach for MXene-AuNBPs based on differences in charge transfer. The original Figure 3c and subsequent information were not clear in the original image. We have enlarged important information in the new Figure 4. Additionally, we have annotated the adsorption energy in the figure legend and the corresponding text.

Due to the relatively low Raman scattering cross-section of 6MNA, the experimentally detected Raman signal was weak, making it unsuitable for comparisons with PATP and 4MBA; therefore, we did not calculate the adsorption energy for 6MNA in this study. The spectral data in the new Figure 4 are clearer. However, since the two-dimensional materials primarily enhance the Raman signal through chemical mechanisms, the signal intensity obtained for the same substance is very low and does not effectively convey trends. To differentiate the changes in SERS signals resulting from charge transfer differences when detecting the same substance, the original Figure 3f (now Figure 4d) only shows different two-dimensional materials detecting PATP.

To highlight the differences in SERS data due to charge transfer, both types of two-dimensional materials are connected to AuNBPs. The SERS performance provided by the

hotspots of AuNBPs makes the differences in charge transfer more pronounced.

When a molecule adsorbs onto the surface of two-dimensional materials, the adsorption behavior affects charge transfer. Lower adsorption energies typically indicate stronger interactions and a more stable adsorption process. When the adsorption energy is low, charge transfer between the adsorbate and a greater amount of charge may be transferred. [arXiv preprint arXiv:2503.17381 (2025); *2D Materials* **5**, 031012 (2018).] Thus, lower adsorption energies usually lead to increased charge transfer, enhancing the SERS performance. To better illustrate the relationship between the two, we have incorporated this content into the main text.

Why was the PATP functionalized surface chosen for detection of TNT?

Response: Thanks for your question. We chose PATP-functionalized MXene-AuNBPs for TNT detection because PATP can interact with TNT. The new Figure 5b illustrates the interaction between PATP and TNT molecules.

The new Fig 5 b. Three interaction mechanisms between PATP and TNT.

Additionally, we have described these interaction mechanisms in the main text as “These molecules interact with nitro compounds, enabling their capture. Taking PATP as an example, it can interact with TNT through three primary mechanisms to achieve TNT capture, as illustrated in Figure 5b. (1) Meisenheimer Complex Formation: PATP forms a Meisenheimer complex with TNT through specific chemical interactions. (2) TNT anions and PATP cations interact via electrostatic ion-pair interactions. (3) π - π Stacking Interaction: TNT acts as a π -acceptor while PATP serves as a π -donor, enabling π - π stacking interactions.”.

On line 337 the authors mention the RSD was 9.82%. What is the average value of the signal? Is the signal with and without Au NBPs correspond to the magnitude of the field

enhancement ⁴ in the hotspots due to configurations shown in Fig. 3a. Is that the source of RSD?

Response: Thank you for your suggestions regarding the RSD values. The results for this section are presented in the revised supplementary information, Fig. 13s. It can be observed that the average intensity of the detected 75 Raman spectra at 1350 cm⁻¹ is 2993. To facilitate a quicker understanding of this average value, the relevant content has been added to the discussion.

In this section, we employed the relative standard deviation (RSD) to assess the variability in the detected Raman signal intensity. A smaller RSD indicates less data variability, suggesting better uniformity in the SERS substrate signal. Therefore, we mainly used the same substrate to detect TNT. To evaluate the spectral intensity at different positions on the substrate, we selected 75 random points for testing. As mentioned by the reviewer, different testing points are indeed influenced by variations in the hotspot distribution of the electromagnetic field shown in Fig. 3a. This fluctuation in signal leads to the corresponding RSD values. We have incorporated the following content into the main text: "Supplementary Fig. 13s further plots the intensity of the TNT characteristic peak at 1350 cm⁻¹ across the 75 points, with an average value of 2993 and a relative standard deviation (RSD) of 9.82%, which is affected by the differences in hotspot distribution at the testing points."

Line 347, "Raman spectra of TNT, 4-NT, and 2,4-DNPA detected at 60°C using the Mo₂C MXene-AuNBPs-PATP substrate" Why was Mo₂C chosen over Ti₂C₂ that had higher signal in Fig. 3?

Response: We thank the reviewer for the question. In our actual experiments, both types of substrates were used for testing. However, to demonstrate the detection capability of a single substrate for nitro compounds, we selected the MXene-AuNBPs-PATP substrate for display. Due to layout issues with the figures, this may have caused some confusion. We have updated the manuscript to include the gas detection experiments using the MXene-AuNBPs-PATP substrate in the new Figure 5 and presented the overall detection results of the SD-SERS arrays in the new Figure 6. This change will help readers more easily distinguish between the single substrate and the SD-SERS arrays.

Line 228 has a typo, maybe a missing reference at the end.

Response: We appreciate the reviewer's comments. We have revised this section to clarify: "When the detection molecules adsorb onto the MXene surface, charge transfer between the metal atoms on the MXene and the molecules enhances the Raman signal through chemical enhancement."

Figure 5 is very busy and hard to read. The values for accuracy, precision are hard to differentiate on the very small figure as well as the comparison of the ML methods, for example.

Response: We would like to thank the reviewer once again for the feedback regarding Figure 5. To address the concerns related to the figure being busy and hard to read, we have split Figure 5 into new Figures 7 and 8 shown as follow. In these new figures, the relevant content has been reorganized and enlarged to enhance clarity and readability.

The new Fig. 7 | **ML-assisted classification of TNT and 2,4-DNPA based on SD-SERS arrays.** a. Classification model based on Machine Learning methods. b. Illustration of data processing for single unit and splicing spectra. c. Results of distinguishing the two gases using Random Forest based on confusion matrices by 10-fold cross-validation. d. (i) The RF classification performance of TNT and 2,4-DNPA gases at 25°C using different models and (ii) the classification accuracy results of the different ML methods for the prediction set of TNT and 2,4-DNPA gases at 25°C.

The new Fig 8| **ML-assisted classification of different concentrations of TNT based on SD-SERS arrays.** a. The RF confusion matrix for the classification of different concentrations of TNT gases using different models. b. The RF classification accuracy of different concentrations of TNT using different models. c. The classification accuracy results of the different ML methods for the prediction set of different concentrations of TNT gases.

Reviewer #2 (Remarks to the Author):

Referee report: SERS nose arrays based on a signal differentiation approach for TNT gas detection

MS No: COMMSCHEM-25-0306-T

Decision: Major revision

1. The authors highlight the SERS array as a key advantage, but it is unclear whether the full array is needed for TNT detection or if a single element suffices. This important detail is missing from the abstract and conclusion and should be clarified for better understanding of the sensor's practical application.

Response: We appreciate the reviewer's suggestions. SD-SERS arrays indeed exhibit outstanding performance in classifying substances with highly overlapping spectra and in

detecting substance concentration. To further highlight this key point in our paper, we have made several revisions.

Firstly, we have reorganized the figures in the paper, which we have elaborated on in our response to Reviewer #1. In brief, we have restructured the original Figure 4 into new Figures 5 and 6, which correspond to the detection effects of a single substrate and the results of the SD-SERS arrays, respectively. Additionally, we have included a flowchart in new Figure 6a shown as follow that illustrates the process of detecting TNT using SD-SERS arrays, emphasizing the method of utilizing the entire array for TNT detection. Furthermore, while analyzing the spectral intensity data for the simultaneous detection of TNT and 2,4-DNPA with the SD-SERS arrays (new Figure 6d(iii)), we have further highlighted the capability of SD-SERS arrays to acquire differentiated information by stating: "These findings demonstrate that SD-SERS arrays leverage differences in chemical enhancement and adsorption capabilities to provide multidimensional spectral features, enabling the differentiation of highly overlapping spectra for TNT and 2,4-DNPA, which is difficult to obtain from Raman spectra using a single substrate."

The new Fig 6

Secondly, we have refined the ML-assisted classification method of SD-SERS arrays

originally shown in Figure 5 into new Figure 7, which corresponds to the classification capability of the SD-SERS arrays for different detected substances, and new Figure 8, which corresponds to the classification capability of the SD-SERS arrays for the same substance at different concentrations. By comparing a single entry in the confusion matrix with the overall classification of the SD-SERS arrays, we further highlight the application potential of SD-SERS arrays in substance classification and concentration prediction.

Finally, we have added a comparison of the performance between single substrates and array substrates in the abstract, introduction, and conclusion, further emphasizing the practical application potential of array sensors, such as “By exploiting the charge transfer differences between various MXene structures and the adsorption capability variations introduced by the monolayers, unique Raman spectral intensity information can be obtained, which were difficult to obtain from Raman spectra using a single substrate. Through the differentiated information obtained from SD-SERS arrays, further integration with machine learning algorithms demonstrates the high accuracy of SD-SERS arrays in classifying TNT and structurally similar 2,4-DNPA, as well as in distinguishing between gases at different concentrations.” in abstract and “According to the experimental results, single substrates face limitations in sensitivity and selectivity when dealing with highly overlapping spectra. This makes it difficult to accurately distinguish between different substances or their concentrations in complex mixtures. In contrast, the SD-SERS array enhances differential recognition between similar compounds through its multidimensional data acquisition capabilities. In this study, the SD-SERS array demonstrated higher classification accuracy and stronger concentration prediction ability.” in conclusion.

2. PTAP, 4MBA, or similar functional molecules are already adsorbed onto a metal surface via physical interactions, what percentage of their functional groups or active sites remains available for TNT detection? Have the authors quantified or estimated the degree of surface accessibility post-adsorption, and how does this affect the sensitivity and reliability of the detection mechanism?

Thank you for the reviewer's question. The functionalized molecules used in this study, as shown in Flowchart 1, all contain thiol groups. During the self-assembly process of these capture molecules onto the MXene-AuNBPs, the thiol groups form Au-S bonds as well as Ti-S and Mo-S bonds with the MXene-AuNBPs. The interactions between the capture molecules and TNT include “(1) Meisenheimer Complex Formation: PATP forms a Meisenheimer complex with TNT through specific chemical interactions. (2) TNT anions

and PATP cations interact via electrostatic ion-pair interactions. (3) π - π Stacking Interaction: TNT acts as a π -acceptor while PATP serves as a π -donor, enabling π - π stacking interactions. For 4MBA, its carboxylic acid group interacts with TNT's nitro group through hydrogen bonding, acting as a Lewis acid-base pair. Similarly, 6MNA shares structural similarities with 4MBA and interacts with TNT in a similar manner." The single molecular layer used in this study reacts with the electron-accepting nitro groups of nitro compounds through the aromatic ring, amino, and carboxylic acid groups. Theoretically, most active sites of the capture molecules modified on the MXene-AuNBPs' surface can be utilized for TNT adsorption. We will incorporate this discussion into the main text.

Regarding the degree of surface accessibility post-adsorption, we estimated the distribution of captured molecules. Specifically, based on the preparation method of MXene-AuNBPs-PATP (4MBA or 6MNA), the mass of 200 μ L of MXene (1 mg/mL) is 0.2 mg. Due to the tight connection of AuNBPs on the surface of MXene-AuNBPs, we estimated the surface area of MXene to represent the surface area of MXene-AuNBPs. To simplify the calculation, we used the surface area of MXene instead of calculating the surface area of MXene-AuNBPs. According to [Applied Surface Science 473 (2019) 409–418], the specific surface area of few-layer MXene is 69.11 m^2/g , leading us to estimate the surface area of MXene-AuNBPs to be approximately 0.0138 m^2 .

The quantity of captured molecules on the surface of MXene-AuNBPs can be calculated as $n = V/C \cdot N_A$, where V is the volume of the solution with 40 μ L, C is the molar concentration of the captured molecules with 10^{-6}M , and N_A is Avogadro's number. This results in a density of captured molecules of $(1.739 \cdot 10^{-3} \text{molecules}/\text{nm}^2)$. For the AuNBPs (height 53 nm, width 20 nm), its surface area is approximately 2000 nm^2 . Thus, at this concentration, each AuNBP can bind about 2.4 captured molecules through Au-S bonds. Regions not covered by the gold nano-dumbbell allow captured molecules to attach to the MXene surface. This facilitates the effective adsorption of TNT on the substrate surface.

As shown in the results of Figure 5c(i), the detection performance of the substrate improved significantly after modification with the captured molecule PATP, increasing from 734.75 to 3349.62, or by a factor of 4.56. This demonstrates that the modified captured molecules can adsorb target substances such as TNT onto the surface of MXene-AuNBPs-PATP (4MBA or 6MNA), enhancing its detection capability.

3. Although the metal substrate and the laser excitation wavelength (785 nm) are in resonance, TNT itself does not exhibit significant electronic absorption near this wavelength. Since the interaction with TNT is mediated via the attached analyte molecule, could the authors clarify how effectively TNT molecules are localized within the plasmonic 'hot spots' of the metal nanostructures? Additionally, it would be helpful to understand whether the spatial distribution and availability of TNT in these hot spots have been quantified or considered in evaluating the sensitivity and reproducibility of the detection.

Response: Thank you for the reviewer 's question regarding "how effectively TNT molecules are localized within the plasmonic 'hot spots' of the metal nanostructures." Based on the estimation of the density of the captured molecules in the previous question, we found that when using 40 μL of 10^{-6} M capture molecules to modify MXene-AuNBPs, each AuNBPs can connect with approximately 2.4 capture molecules. Any unbound capture molecules will attach to the MXene surface. The high-index facets of the AuNBPs typically have a higher surface formation energy, making these capture molecules preferentially bind to the high-index surfaces during the connection process.[Chemical Engineering Journal, 2024, 485: 150045] Consequently, capture molecules mainly localize at the tips and edges of AuNBPs, which are the SERS hot spot regions. The interactions between the modified capture molecules and TNT result in the adsorption and accumulation of TNT molecules near these hot spots.

The spatial distribution of capture molecules on the substrate surface determines the effective utilization of substrate hot spots. To validate this, we compared the Raman signals from the same concentration of TNT gas detected after modifying MXene-AuNBPs with different concentrations (10^{-6} M and 10^{-5} M PATP) of capture molecules, as shown in Figure 11s. According to the calculations of the capture molecule density, 40 μL of 10^{-6} M capture molecules mainly localized in the hot spots of the AuNBPs and on the MXene surface. In contrast, the modification with 40 μL of 10^{-5} M capture molecules increased the density tenfold, resulting in a more dispersed distribution of PATP on the surface of the AuNBPs. This dispersion may cause the interacting TNT molecules to be located further away from the hot spots.

Fig. 1s| Raman spectra of Mo₂C MXene/AuNBPs composite SERS substrates modified with 10⁻⁵ M and 10⁻⁶ M capturers before (b and d) and after (a and c) detecting 15 ppb TNT.

Although a higher concentration of modified capture molecules may capture more TNT molecules, those located farther from the hot spots yield lower detected signals. Furthermore, the increased Raman signals from excess capture molecules may interfere with the detection of TNT signals, reducing the overall intensity of the detected TNT signals. As shown in Figure 11s, the TNT signal intensity detected from MXene-AuNBPs-10⁻⁶ M PATP exceeds that from MXene-AuNBPs-10⁻⁵ M PATP by 3.6 times. This indicates that by optimizing the spatial distribution of detection molecules at the substrate hot spots, we achieve superior detection performance. Relevant content has been included in the supporting information and the manuscript.

4. The term 'physical coupling methods' is not standard or commonly used in the context of nanoparticle synthesis, particularly for Ag and Au nanoparticles. It would be helpful if the authors could clarify what they mean by this term or consider using more widely accepted terminology such as 'physical synthesis methods' or specify the particular techniques involved (e.g., laser ablation, evaporation-condensation). Clear definitions will improve the clarity and scientific rigor of the manuscript.

Response: Thank you for the reviewer's suggestions. "Physical coupling methods" is not suitable here. A more acceptable term, "self-assembly method assisted by electrostatic attraction," has been used instead.

5. Although the authors mention control experiments for selectivity, the discussion and experimental details provided are unclear and somewhat confusing. The manuscript lacks

a systematic and well-structured presentation of control data that convincingly demonstrates the sensor's ability to distinguish TNT from structurally similar compounds or potential interferents. For example, competitive binding experiments involving TNT and related analytes would provide stronger evidence of selectivity. Before performing comparisons with machine learning data obtained from the electronic nose, it is important to first establish a clear and robust baseline through well-designed control and selectivity experiments.

Response: We appreciate the reviewer's questions. The classification method used in this study focuses on the detection of gas molecules that are structurally similar and have overlapping spectra. The selective signal differences from various substrates in the SERS nose provide valuable data for distinguishing structurally similar substances. To highlight the unique signal characteristics associated with different analytes, we experimentally demonstrate the classification capabilities of the SERS nose compared to a single substrate. Additionally, competitive binding experiments with TNT and related analytes typically involve substances with differing Raman shift characteristics, which is why this experiment was not conducted in our study.

6. The authors describe an approach where the SERS substrate is incubated with gas-phase analytes and then removed for Raman measurement. While this method may be effective for controlled lab studies, it raises questions about its applicability for real-time or in situ detection. The manuscript would benefit from a discussion on whether the setup can be adapted for continuous or portable SERS-based sensing, as real-time detection is crucial for practical, field-deployable applications.

Response: Although this study was conducted in a controlled laboratory setting, we did not explore practical applications in real-world environments due to the pathogenic nature of the analytes. However, the entire detection process is suitable for real-time monitoring or in situ detection. Therefore, we have discussed the portability of the SERS nose and the feasibility of real-time detection in the conclusion.

Further comments below require additional attention:

7. Line 37: Different style of citation. Please check. It even starts with citation 2, afterwards style is changed. Is the citation [2] the same as 2?

Response: Thank you to the reviewer for pointing out the issue. There was indeed an error

in the wording of [2]. We have corrected the relevant content in the paper and checked the entire manuscript to ensure that all citations are accurate.

8. Line 58: Check misspelling, such as 'senors' instead of sensors.

Response: We thanks again for the reviewers' commands. The relevant content has been revised,

9. Line 81: The example of food safety needs citation(s).

Response: We have added the citation for food safety [Trends in Food Science & Technology, 2021, 112: 225-240].

10. Line 93-95: Why is electromagnetic enhancement no key factor?

Response: We appreciate the reviewer's question. The intention here is to highlight two factors contributing to the differentiated signals of SD-SERS arrays. Physical enhancement of signal strength does not lead to signal differentiation. However, since physical enhancement can amplify the intensity of differentiated signals and is a key factor in SERS enhancement, it is necessary to clarify this. To avoid confusion, we consider electromagnetic enhancement as another factor influencing the signals of SD-SERS arrays. Relevant content will be added to the appropriate sections.

11. Line 105-107: What is a physical enhancement? Please clarify the role of EM enhancement.

Response: We appreciate the reviewer's suggestion. We have removed the term "physical enhancement" in paper and emphasized the role of electromagnetic (EM) enhancement.

12. Line 112/113: "the physical enhancement structure of the SD-SERS arrays should exhibit superior electromagnetic enhancement hotspots". Why 'should'? The arrays exhibit EM hotspots!

Response: Thank you for the commend. We delete the "should".

13. Line 114: "physical enhancement "hotspots". Better EM enhancement than physical enhancement.

Response: Thank you for your guidance. The corresponding content has been modified as requested.

14. Line 118: "with a tip-to-tip distance of 1.5 nm in all three cases". Is this a realistic value for fabrication of these structures?

Response: Thank you for the reviewer's question. The 1.5 nm value is meant to illustrate

the electromagnetic enhancement hotspot performance of the gold nanostructure, rather than being an actual measurement.

15. Line 131: “[...] and chemical enhancement properties were obtained”. Do they show no EM enhancement?

Response: Thank you for your suggestion. To avoid ambiguity, we have revised this section as “By self-assembling three types of monolayers onto the surfaces of the two structures, we obtained six SERS substrates with similar electromagnetic enhancement features but different adsorption capacities and chemical enhancement properties. These six substrates collectively form the SD-SERS arrays.”

16. Line 161: CTAB instead of CTAC?

Response: Thank you for the reviewer's comments. The synthesis of AuNBPs mainly has two steps. First, gold seeds were prepared, and CTAC is used as a surfactant in this step. Then, the AuNBPs were synthesized assisted by Au seeds, and CTAB is used as a surfactant in this step. The use of the term CTAB here is correct. We've also checked the whole paper to make sure the descriptions of surfactants are correct in every part.

17. Line 171/172: “This resonance enhances the intrinsic SERS capability of the substrate, thereby improving its overall performance.” Misleading, please improve the sentence.

Response: Thank you for your suggestion. The corresponding content has been removed.

18. Line 229/230: Are the “head-to-head, head-to-shoulder, and shoulder-to-shoulder configurations” realistic for the fabricated SERS substrates?

Response: Thank you for the reviewer's questions. The SERS effect of the SD-SERS array substrate is primarily generated through electromagnetic enhancement hotspots. In the actual substrate, the randomness of the arrangement of AuNBPs allows for the formation of the three mentioned configurations on the substrate surface. The calculations indicate that these hotspots all exhibit high electromagnetic enhancement characteristics. This contributes to the high sensitivity of the SERS substrate presented in this study. At the same time, discussing these three types of hotspots can further explain the optimization of the SERS performance of the substrate mentioned later in the text. Therefore, addressing the significance of these three hotspots is essential.

19. Line 234: beginning of the line. Is there a reference missing?

Response: We appreciate reviewers' suggestion. The corresponding content has been

removed.

20. Line 249: What about stronger EM enhancement in those cases the analyte is closer to the surface due to attractive interaction?

Response: Thank you for your questions. In the previous response, the authors suggest that by optimizing the substrate parameters, the analytes will be attracted and enriched near the hotspots of the substrate. Given that these hotspots exhibit high electromagnetic enhancement properties, it can be concluded that the electromagnetic enhancement capability of the substrate is more effectively utilized.

21. Line 288 sentence is incomplete "According to (), MXene also have smaller SERS performance than Ti_3C_2 "

Response: Thank you for your suggestion. The corresponding content has been removed.

22. In Fig. 3e and 3f, two different styles to highlight the PATP modes were used and in one case 2 modes are marked and in the other only 1 mode. Why?

Response: We appreciate the reviewers for their careful examination of the manuscript and their valuable feedback. To avoid confusion, we standardized the labeling style. In Figure 3e, we only labeled one mode because the signals of other modes were difficult to observe in the spectra. In Figure 3f, two modes are labeled due to the enhanced signals of the other modes through the SERS effect of MXene-AuNBPs.

23. Fig. 3e and 3f is lacking from visibility. It is too small. Spectral features are barely visible. Please improve.

Response: Thank you for your suggestion. We divided the original Figure 3 into two parts. The new Figure 3 focuses on the coupling hotspot effect between AuNBPs and the electromagnetic field enhancement in MXene-AuNBPs. The new Figure 4 describes the charge transfer behavior between different MXene materials and adsorbed molecules, and experimentally validates that the differences in adsorption behavior can lead to variations in the intensity of the enhanced Raman signals from different MXene-AuNBPs. The size of the figure has been also adjusted for improved clarity.

24. Fig. 4: To which vibration the peak around 1440 cm^{-1} is assigned?

Response: Thanks again for your question. For the peak at around 1440 cm^{-1} is assigned to vibration of C-S for PATP. It can be observed that certain interactions occur between PATP molecules, resulting in charge transfer.

25. Line 301: How the formation of a monolayer is confirmed?

Response: We appreciate the reviewer's suggestions. We have calculated the density distribution of the captured molecules on the substrate surface. The results show that the detected molecule density is below 1 molecular/nm², indicating that the captured molecules can be considered as a monolayer structure.

26. Line 319/320: Is a change of the signal observed for PATP after interaction with TNT and is this useable for a concentration-dependent measurement of TNT or similar molecules? Or is the analysis only based on the specific TNT Raman mode?

Response: We are grateful for the reviewer's question. Some experimental results indicate that PATP [Anal Chem, 2011, 83(18): 6913-7] or p-aminobenzenethiol (PABT) [Analytical chemistry, 2014, 86(7): 3338-3345] can interact with TNT molecules through charge transfer to form a Meisenheimer complex. This complex causes chemical enhancement of the SERS substrate, which in turn increases the signal intensity of the Raman marker PATP or PABT. As the concentration of TNT molecules increases, the Raman signal of PATP or PABT becomes stronger, demonstrating a positive correlation between the two. Therefore, the signal intensity of PATP or PABT can be used to indirectly detect the concentration of TNT molecules.

However, similar phenomena were not sufficiently observed in the experiments of this study, which may be due to the relatively smaller concentration of PATP modifications employed. Additionally, the aim of this study is to achieve direct detection of TNT by monitoring the characteristic peak of TNT. Therefore, we have not explored this method more deeply. Nevertheless, both the underlying principle and results from other experiments indicate that this approach is indeed feasible.

27. Line 346: The concentration values of TNT need to be related with relevant concentrations in real application scenarios.

Response: We are grateful for the reviewer's suggestion. We have conducted the relevant calculations in the supporting information, as can be seen in Table 4s. We will incorporate this explanation into the main text. Therefore, the following sections will discuss the TNT gas concentrations under the experimental conditions that generate gas at specific temperatures. This is primarily aimed at confirming the analytical capability of SD-SERS arrays in detecting different gases produced in the same environment during actual

detection conditions.

28. Line 347/348: Please mention here the band assignment.

Response: We are grateful for the reviewer's suggestion. The characteristic peak mentioned in paper are attributed to the symmetric stretching vibration of the nitro group (NO₂). We have revised the corresponding section.

29. Line 389-391: How to deal with the significant differences in Raman intensity across the various SERS substrates? How to define the measurement procedure that stable and comparable measurement conditions are guaranteed?

Response: The differences in signal responses when detecting the same substance on various substrates arise from the distinct adsorption and charge transfer capabilities of each substrate. This differentiated signal response imparts unique characteristics to different substances. As a result, we can define a method to achieve high accuracy in discrimination by utilizing the varied signals from different substrates, even when a single substrate cannot distinguish between substances with similar characteristic peaks and intensities. This explains why the signals obtained from SD-SERS arrays, when analyzed with machine learning, yield a higher accuracy than those from a single substrate model. During the detection process, the experimental procedures are strictly followed as described in the paper to ensure consistency throughout the entire experiment.

30. Line 428: How many spectra per concentration? How many replicates? It should be at least 3 replicates tested on 3 different SERS substrates per concentration value for statistical relevance.

Response: Thank you for the reviewer's questions. We apologize for not clearly describing the signal acquisition conditions in the paper. To train our machine learning model, we conducted gas detection at different temperatures. We selected 60 random points on each individual substrate of the SD-SERS arrays for detection and then prepared new SD-SERS arrays for further testing. This process was repeated three times, resulting in 180 spectral data points from each individual substrate. Overall, we collected a total of 16,200 signal data points from the SD-SERS arrays across various gases and concentrations. We have added this information to the experimental section.

31. Line 476: "[...] but performed poorly in concentration classification". Does that mean that a quantification is not achieved? What is the required detection sensitivity for the target

analytes? And is this concentration value identified?

Response: Thank you for the reviewer's question. This does not imply that unit e has not achieved quantification. The conclusion is based on our comparison of the quantitative results of the SD-SERS arrays as a whole with those of a single substrate. From the experimental data, we can see that under multiple repetitions, the classification results for unit e remain below 90%, while the classification results for the SD-SERS arrays exceed 99%. This demonstrates that the quantification of TNT using unit e is "poorly". The concentrations we use in our experiments are identified.